# Repair of naturally occurring mismatches can induce mutations in flanking DNA

Jia Chen, Brendan F Miller, Anthony V Furano*

Section on Genomic Structure and Function, Laboratory of Cell and Molecular Biology, National Institute of Diabetes and Digestive and Kidney Diseases, National Institutes of Health, Bethesda, United States

**Abstract** 'Normal' genomic DNA contains hundreds of mismatches that are generated daily by the spontaneous deamination of C (U/G) and methyl-C (T/G). Thus, a mutagenic effect of their repair could constitute a serious genetic burden. We show here that while mismatches introduced into human cells on an SV40-based episome were invariably repaired, this process induced mutations in flanking DNA at a significantly higher rate than no mismatch controls. Most mutations involved the C of TpC, the substrate of some single strand-specific APOBEC cytidine deaminases, similar to the mutations that can typify the 'mutator phenotype' of numerous tumors. siRNA knockdowns and chromatin immunoprecipitation showed that TpC preferring APOBECs mediate the mutagenesis, and siRNA knockdowns showed that both the base excision and mismatch repair pathways are involved. That naturally occurring mispairs can be converted to mutators, represents an heretofore unsuspected source of genetic changes that could underlie disease, aging, and evolutionary change.

*For correspondence: avf@helix.nih.gov

Competing interests: The authors declare that no competing interests exist.

## Introduction

Species survival depends on the faithful replication of genetic information which is monitored and maintained by a number of complex and interacting DNA repair pathways (*Modrich, 2006*; *Cannavo et al., 2007*; *Cortázar et al., 2007*; *Hsieh and Yamane, 2008*; *Kunz et al., 2009*; *Robertson, 2009*; *Liu et al., 2010*; *Jacobs and Schar, 2012*; *Jiricny, 2013*). Continual DNA repair is required to correct the thousands of genetic lesions that occur daily due to just the inherent chemical lability of DNA (*Atamna et al., 2000*; *Barnes and Lindahl, 2004*). For example, the susceptibility of C (and its methylated derivative) to spontaneous hydrolytic deamination daily generates hundreds of U/G and T/G mismatches respectively, (*Barnes and Lindahl, 2004*) and could explain why C is the most frequent source of single nucleotide substitutions in mammals (*Hwang and Green, 2004*).

Error-free base excision repair (BER) can correct the naturally occurring U/G and T/G mismatches (reviewed in *Cortázar et al., 2007*; *Hegde et al., 2008*; *Robertson, 2009*; *Jacobs and Schar, 2012*). The basic reaction involves removal of the base that is paired with G by anyone of several glycosylases to generate an abasic site that is cleaved on its 5′ side by the APE1 endonuclease. The resulting 3′ end is extended by insertion of dCMP by the high fidelity polymerase β coincident with its hydrolysis of the 5′-phosphodeoxyribose that had been generated by the glycosylase. This step is followed by sealing the resulting single stranded break (SSB) by DNA ligase III with the assistance of the scaffolding protein XRCC1.

However, in lymphoid (B) cells the U/Gs that are generated by activation-induced cytidine deaminase (AID, a member of the AID/apolipoprotein B mRNA editing enzyme, catalytic polypeptide-like (APOBEC) family of cytidine deaminases, *Conticello, 2008*) on transient single-stranded DNA regions produced during transcription, are prone to several mutagenic processes that enhance diversification of immunoglobulins (somatic hypermutation, SHM, *Martomo and Gearhart, 2006*; *Teng and Papavasiliou,*

**eLife digest** The inherent chemical instability of the four bases that are found in DNA leads to our genetic material being damaged on a daily basis. The sequence of these bases codes the genetic instructions necessary for all cellular functions, so damaged bases must be efficiently recognized and accurately repaired. The base excision repair pathway carries out these functions.

However, there are some circumstances in which random changes to the genetic code can be beneficial. In immune cells, for example, these changes enhance the diversity of antibodies generated to fight bacteria and viruses. In immune cells, a second repair pathway—the mismatch repair pathway—hijacks the base excision repair pathway. This gives enzymes belonging to the APOBEC family access to the DNA that is undergoing repair, and these enzymes change cytosine bases to uracil bases. Subsequent processing steps can lead to different bases substituted for the original cytosine. The recent discovery that APOBEC enzymes are abundant in other types of cells raised the possibility that these enzymes could be significant source of mutations in the DNA of cells where such mutations are not welcome.

To explore this possibility Chen et al. deliberately introduced a number of mutations (that are normally repaired by the base excision repair pathway) into non-immune human cells and observed what happened. The mutations were repaired, but the number of mutations in neighboring bases increased by a statistically significant amount. In particular, most of these mutations involved a cytosine base that was preceded by a thymine base. Chen et al. also showed that both APOBEC and the mismatch repair pathway are involved, as is the case for the mutations caused by APOBEC enzymes in immune cells. Similar APOBEC mutations are known to be involved in cancer.

The model system developed by Chen et al. not only shows that normally error-free DNA repair can be involved in generating these mutations, but also used to obtain a better understanding of these processes and thereby provide new insights in cancer biology.

*2007*; *Peled et al., 2008*). One of these involves processing U/G mismatches, or BER products thereof, by a non-canonical application of the mismatch repair (MMR) pathway. Normally, MMR (*Modrich, 2006*; *Hsieh and Yamane, 2008*; *Jiricny, 2013*) is a high fidelity process that operates post-replicatively on the nascent DNA strand to remove mismatches that have escaped the proof-reading activity of high fidelity replicative DNA polymerases. Essential components of this pathway include the heterodimer MutSα (MSH2 and MSH6), which recognizes mismatches, the heterodimer MutLα (MLH1 and PMS2), which accesses the mismatch-containing nascent strand in a reaction mediated by proliferating cell nuclear antigen (PCNA, a multipurpose replication clamp, e.g., *Moldovan et al., 2007*; *Lee and Myung, 2008*). PCNA also activates a latent endonuclease in MutLα (*Kadyrov et al., 2006*; *Pluciennik et al., 2010*), which provides entry points for the EXO1 nuclease that excises the mismatch-containing nascent strand to expose the replication template for re-copying by a high fidelity DNA polymerase, such as polymerase δ.

In B cells, non-canonical MMR can expose single stranded regions at U/G-containing sites unrelated to DNA replication that can serve as a template for DNA repair. However, the high fidelity DNA polymerase is replaced by the error-prone polymerase η (mediated by mono-ubiquitylated PCNA). Thus, MMR is subverted to an error-prone process that contributes to SHM. Recently, elements of non-canonical MMR have been recapitulated in vitro (*Schanz et al., 2009*; *Peña-Diaz et al., 2012*), and the latter study also showed that extracts of non-lymphoid mammalian cells can also process U/G mismatches by a non-canonical MMR process. In addition, these non-lymphoid cells when stressed in vivo by the alkylating agent, N-methyl-N′-nitro-N-nitrosoguanidine (MNNG), generated more mutations in MMR proficient than deficient cells, thereby implicating MMR in the mutagenic process but presumably using an AID-independent mechanism (*Hsieh, 2012*).

As DNA homeostasis would seem to require a continual state of DNA repair, its involvement in error-prone processes even at a low frequency would have important implications for the mutational mechanisms that could underlie evolution, aging, and disease. Of interest in this regard is the mechanism that produces the enhanced mutation rate that characterizes certain tumors, which has been termed the 'mutator phenotype' (*Bielas et al., 2006*; *Venkatesan et al., 2006*). Recent examples of

such a mutator phenotype are the high mutation rates of the C of TpCpN (or its complement, G of NpGpA) that can accompany the progression of some cancers (*Nik-Zainal et al., 2012*; *Roberts et al., 2012*). TpC-preferring members of the AID/APOBEC family of C deaminases, particularly APOBEC3B (A3B), mediate these mutations (*Burns et al., 2013*; *Leonard et al., 2013*; *Roberts et al., 2013*; *Taylor et al., 2013*), which can occur in strand-coordinated clusters. These deaminases prefer single stranded DNA and an important issue is how the single stranded APOBEC substrate is generated. Experiments using yeast as a model system showed that the transitory single stranded regions that arise at replication forks or during double strand break repair can accumulate such mutations upon chronic alkylation of DNA (*Roberts et al., 2012*). These mutations were also observed at experimentally introduced double strand breaks coincident with overexpression of AID/APOBEC deaminases (*Taylor et al., 2013*).

Here, we determined directly whether repair of the naturally occurring T/G and U/G mismatches would be mutagenic to flanking DNA in mammalian cells that were not stressed by genotoxic agents. We introduced these mismatches (and other mispairs and lesions) into an SV40-episome that can replicate in human cells. Numerous studies have shown that processing lesions harbored by such episomes (as well as those introduced in the SV40 virion) faithfully captured the DNA repair repertoire and mutational environment of its host (e.g., *Hare and Taylor, 1985*; *Seidman et al., 1985*; *Brown and Jiricny, 1987*; *Choi and Pfeifer, 2005*; *Terai et al., 2010*; *Pathania et al., 2011*; *Qi et al., 2012*). We found that the introduced mispairs were invariably corrected. However, their repair generated mutations in normal flanking DNA at statistically higher rates than the no mismatch control, but only if the episomes were passed through mammalian cells.

Regardless of the lesion, most of the mutations involved the C of TpCpN, and shared other features of the aforementioned mutations that occur in some cancers. siRNA knockdowns showed that the TpC-preferring deaminases, particularly A3B mediated the mutagenic effect, and chromatin immunoprecipitation (ChIP) showed that A3B could access the episome in a mismatch dependent way. siRNA knockdowns also showed that components of both the BER and MMR pathways are involved in generating the single-stranded APOBEC substrate and that two factors that have diverse roles in various aspects of DNA metabolism, ataxia telangiectasia and Rad3-related protein (ATR) and PCNA (*Moldovan et al., 2007*; *Flynn and Zou, 2011*) modulate the mutagenic effect.

The APOBEC deaminases are relatively ubiquitous in various tissues (*Refsland et al., 2010*), as is the potential for generating single-stranded templates from BER processed lesions. These would not only include the mispairs generated from C and methyl-C by their spontaneous deamination as we found here, but potentially also those involving other normal metabolites of methyl-C (e.g., *Guo et al., 2011*; *Wu and Zhang, 2014*) and the thousands of BER substrates that arise daily from other naturally occurring degradative processes that affect DNA (*Atamna et al., 2000*; *Barnes and Lindahl, 2004*). The implications of our findings in light of these issues are discussed.

## Results

### DNA repair can induce mutations in flanking DNA

*Figure 1A* shows the SV40-based episome (shuttle vector) (*Parris and Seidman, 1992*) into which we inserted a mismatch region (MM1). Both strands of MM1 contain two sites for a different pair of single strand restriction enzymes, which facilitate the exchange of either strand with an exact complement to generate a no mismatch (0 MM) control or with an oligonucleotide that would generate a mismatch or other lesions (*Figure 1B*, 'Materials and methods', *Figure 1—figure supplements 1 and 2*). *Supplementary file 1* lists the oligonucleotides used and their corresponding numbers are given in the figures. After passage in mammalian cells we screened the episomes for mutations by blue/white selection in *E. coli* (*Figure 1B*) and determined the DNA sequence of the reporter cassette of the episomes from all the white colonies.

*Figure 2A* shows that the repair of different types of mismatches or lesions - T/G, 5-hydroxymethyl-U (hmU)/G, U/G, or an abasic site opposite a G, (ab)/G – induced significantly more mutations in the reporter region than we found with the 0 MM control. HmU is a byproduct of the enzymatic demethylation of methyl-C (*Bhutani et al., 2011*; *Guo et al., 2011*; *Wu and Zhang, 2014*) and abasic sites are generated during BER (e.g., *Robertson, 2009*; *Jacobs and Schar, 2012*). For convenience, we refer to both mismatches and ab/G sites as lesions. Mutation frequency is the percent

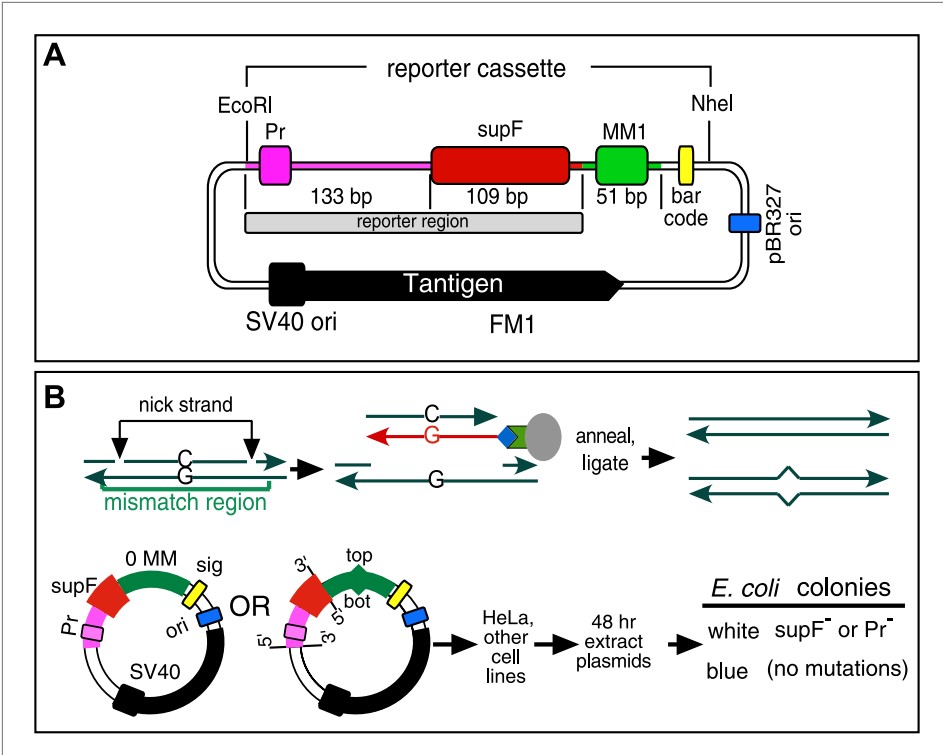

**Figure 1**. Procedure to determine mutagenic effect of DNA repair. (**A**) Episome FM1: purple double line with square, *supF* promoter region (Pr); red rectangle, *supF* gene; gray rectangle, reporter region; green box, mismatch region 1 (MM1); yellow rectangle, bar code; dark blue rectangle, pBR327 origin (ori, FM1 replication in *E. coli*); black square and contiguous heavy black arrow, SV40 origin-promoter and T-antigen coding sequence respectively (FM1 replication in human cells). We refer to this episome as FM1 because the *supF* gene is upstream of the MM1. Not shown, the *Ap^R* gene, which renders *E. coli* resistant to ampicillin. (**B**) Steps to generate lesion-containing vectors and procedure for determining the mutagenic effect of DNA repair in mammalian cells: Mismatch region with nicking sites (vertical arrows) is digested with a single strand restriction enzyme on the top (or bottom) strand, and the nicked strand is removed by hybridization to a 5' biotin (blue diamond) labeled complementary DNA (red). The hybrid is then tethered to a streptavidin (green polygon)-coated magnetic bead (gray oval). The purified gapped episome is reconstituted by ligation to its perfect complement or an oligonucleotide that contains one or more mismatches to generate vectors with a top (or bottom) strand lesion or its corresponding 0 MM control. These vectors are transferred to mammalians cells, harvested after 48 hr and subjected to blue/white screening. The procedure is described in detail in the 'Materials and methods'.

The following figure supplements are available for figure 1:

**Figure supplement 1**. Monitor the gapping and reconstitution of episomes by KpnI digestion.

**Figure supplement 2**. Monitor the presence of mismatch generating oligonucleotides by AatII digestion.

(%) white colonies (per total screened) that contained undeleted episomes. We did not consider deletions because most were missing all or part of the reporter region. These deletions had resulted from our initial method of vector preparation and were essentially eliminated by its subsequent modification (*Figure 2—figure supplement 1* and 'Materials and methods-Vector preparation'). The few percent that persisted were unrelated to either the type or even presence of an introduced DNA lesion (see 'Materials and methods-Data acquisition and analysis'). Finally, no mutated episomes were obtained if they were passed directly into *E. coli* (3 deleted episomes /42,000 colonies screened, results not shown).

The relative mutagenic effect of repairing a given type of lesion on a given strand (top or bottom) was more or less indifferent to its number (up to three), context (i.e., CpG or non-CpG) or position(s) in the MM region (*Figure 2—figure supplement 2*; *Supplementary file 1*). Thus, the mutagenic

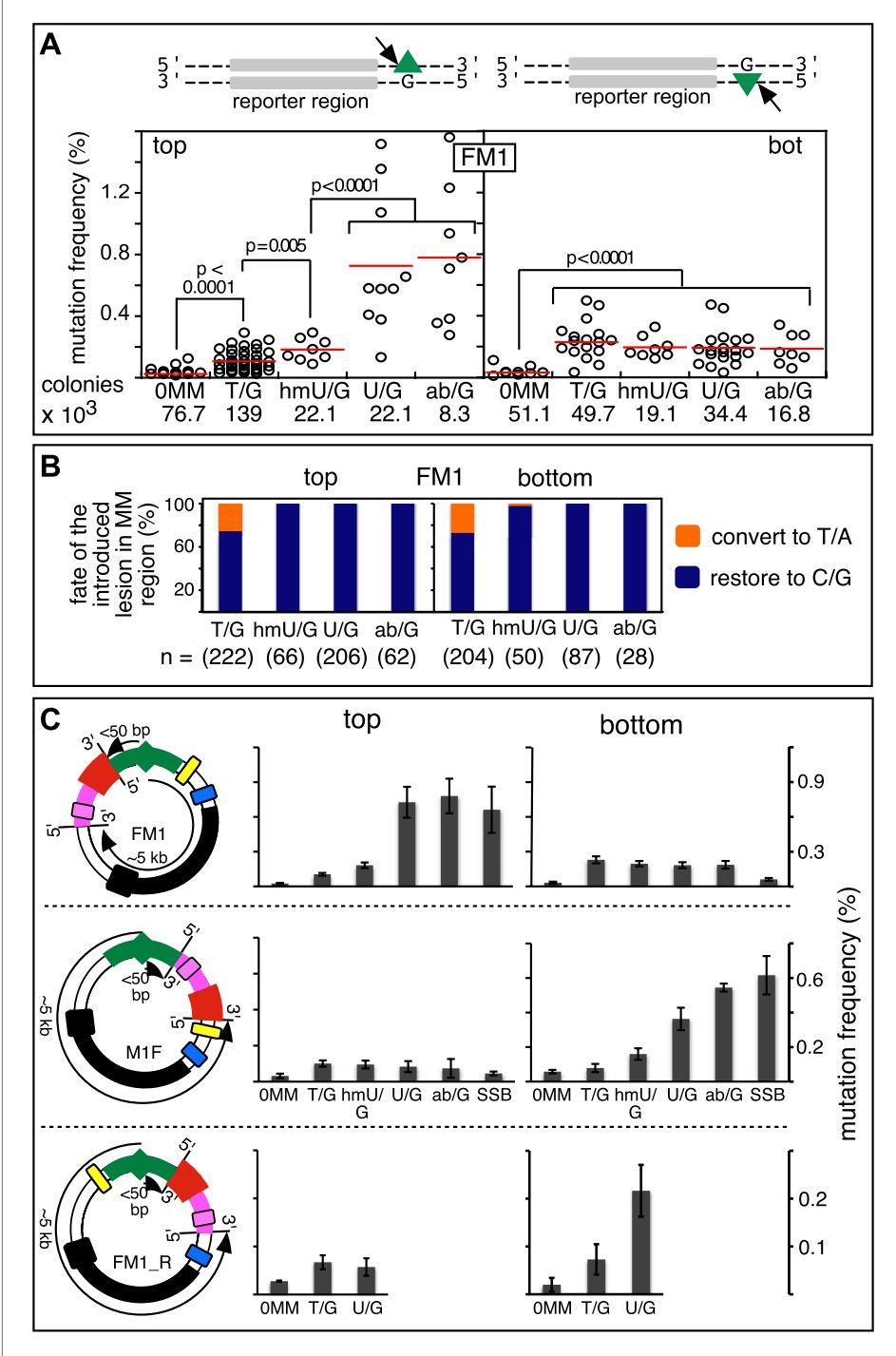

**Figure 2**. Mutagenic effect of DNA repair. (**A**) Dot plots of the repair-induced mutation frequency (number of white colonies with undeleted episomes/total—see 'Materials and methods' and text) as a function of the indicated lesion. Each dot represents a separate trial of a given type of lesion that was present in 1, 2, or in the case of T/G, 3 copies in various positions (and thus sequence contexts) in the mismatch region. See *Figure 2—figure supplement 2*; *Supplementary file 1*. The red horizontal lines indicate the mean, and p values were calculated by the Fisher exact test on pairwise 2 × 2 contingency tables of the total number of non-mutant colonies (blue) and total number of white colonies that contained undeleted episomes. We applied a Bonferroni correction for multiple comparisons by multiplying the p values by the number of comparisons for the top or bottom strand lesions. The diagrams above the dot plots depict the reporter cassette. The green triangle indicates the location of the *Figure 2. Continued on next page*

*Figure 2. Continued*

introduced lesion, and the arrow indicates the 5' single-stranded break that would be generated by BER. (**B**) The fate of the introduced mismatches in the MM region present in the mutated plasmids (pooled according to the type and strand location of the lesions). The percentages of lesions restored to C/G (blue) or converted to T/A (orange) are shown along with the numbers of introduced lesions. (**C**) Configurations of episomes FM1, M1F and FM1_R, and bar graphs of the mutation frequency as a function of the indicated lesion introduced on the top or bottom strand (pooled according to the type and strand location of the lesions). Means (±SEM) are shown from at least two independent experiments. The bar graphs shown for FM1 are the same data as the dot plots in panel **A**. The distances between the lesions and the *cis* 3' end of the reporter are given—the diagrams of the episomes are not to scale.

The following figure supplements are available for figure 2:

**Figure supplement 1**. Minimal manipulation of reconstituted episomes minimizes the generation of deletions.

**Figure supplement 2**. The magnitude of repair-induced mutagenesis differs with the type of lesion but the number or nucleotide context of a given lesion does not materially alter its mutagenic effect.

**Figure supplement 3**. The mutagenic effect is not affected by the C+G content of the mismatch region.

**Figure supplement 4**. The mutagenic effect of single strand break repair supersedes that of T/G repair.

---

effects of repairing the various configurations of T/G, hmU/G and U/G shown in *Figure 2A* can be recapitulated by a given set of these lesions (i.e., 2 T/Gs, 2 hmU/Gs or 2 U/Gs at the same positions in the MM region, the uppermost bar graph of *Figure 2—figure supplement 2*).

The top or bottom strand location of the lesions differentially affected their mutagenic effect: repair of top strand T/Gs generated ~fourfold more mutated episomes than the 0 MM control (respective mean percent mutation frequency 0.107 vs 0.025). HmU/G repair induced ~sevenfold more mutations (0.182%), and U/G or ab/G repair induced ~30-fold more mutations (0.726% and 0.78% respectively) than the 0 MM control. These differences were statistically significant (legend, *Figure 2A*). The bottom strand location reduced the mutagenic effect of repairing U/G or ab/G, but not of T/G or hmU/G. Repair of bottom strand T/G was ~twofold more mutagenic than on the top strand: 0.23% ± 0.013 vs 0.11% ± 0.011 (mean ± standard error, $p < 10^{-7}$, Fisher exact test).

The differences in mutagenesis were not due to differences in repair efficiency. *Figure 2B* shows that, except for T/G, essentially all of the top and bottom strand lesions were restored to C/G. T/G was converted to T/A about 25% of the time. This would result from using the T-containing strand as the repair template (see section, 'The mutational spectrum, sequence context, and fate of the bases mutated in response to DNA repair are consistent with APOBEC-mediated mutagenesis'). The mutated plasmids represented repair of 426 T/G, 116 hmU/G, 293 U/G and 90 ab/G mismatches (combined top and bottom lesions, *Figure 2B*), and the percent restoration to the C/G base pair was: 72.3, 99, 100 and 100 respectively. The values for the restoration of lesions in the non-mutated episomes (isolated from blue colonies), were 75% of 111 T/G, 95.8% of 24 hmU/G, 99% of 105 U/G, and 100% of 7 ab/G mismatches. Thus, the ratio of restoring T/G to C/G or converting it to T/A was independent of repair-induced mutagenesis. Conversion of T/G mismatches to T/A had also been observed with T/G-containing SV40 virion DNA (e.g., *Hare and Taylor, 1985*; *Brown and Jiricny, 1987*). The mutagenic effect was also independent of the G+C content of the mismatch region; 69% G+C for FM1, 39% for FM2, and 53% for FM3 (*Figure 2—figure supplement 3*). Thus, mutagenesis was not affected by the presumed stability of the MM region helix.

## The distance between the lesion and the *cis* 3' end of the reporter region affects mutagenesis

The distance between the *cis* 3' end of the reporter region and the lesion is <50 bp for top-strand, but ~5 kb for bottom-strand lesions in FM1 (top panel *Figure 2C*). This difference accounts for their different mutagenic effects because relocating the MM region upstream of the reporter region (M1F) switches these distances (now <50 bp for bottom—but ~5 kb for top-strand lesions respectively), and it also switches the extent of their respective mutagenic effects (cf. top and middle

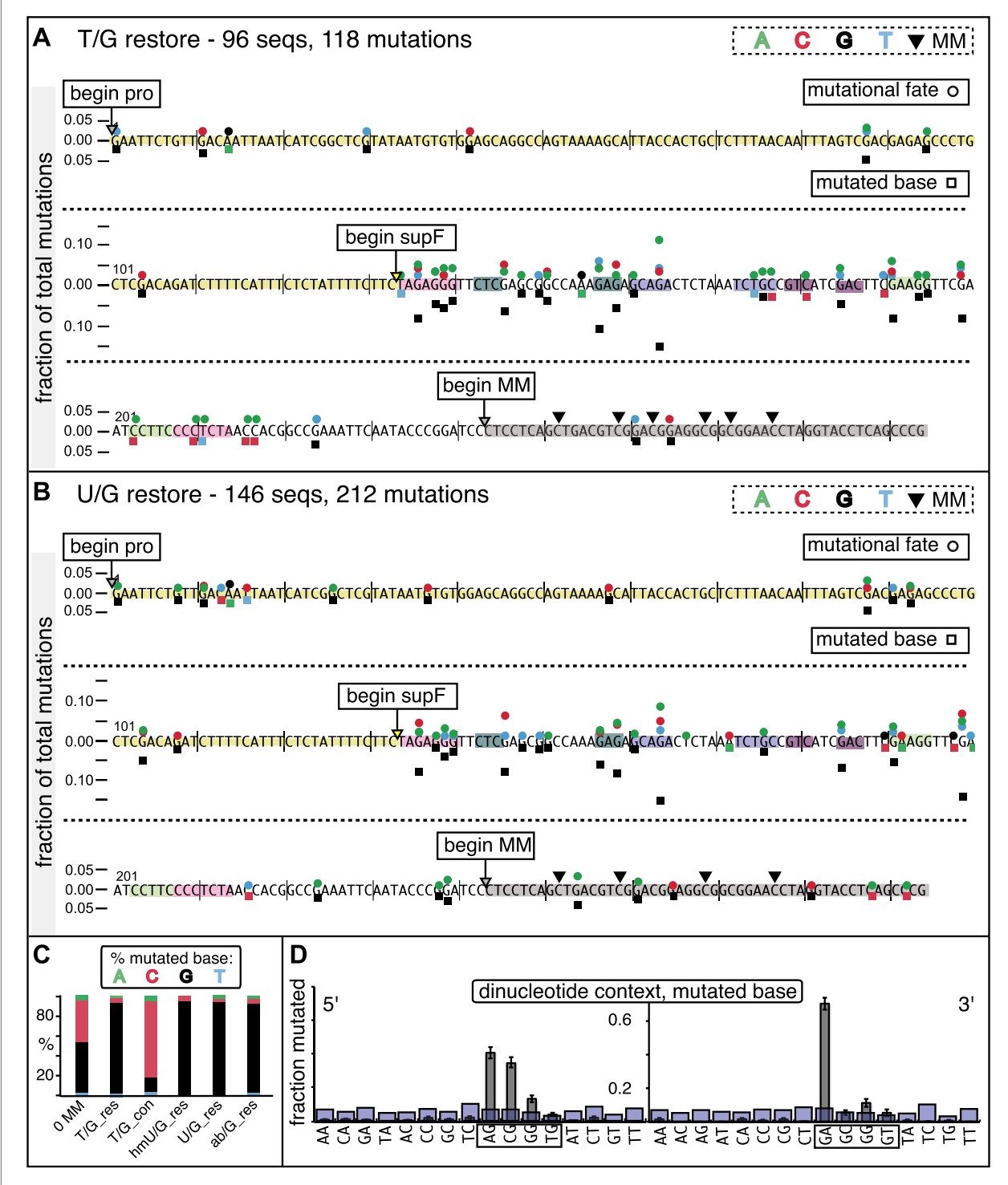

**Figure 3**. Mutational spectrum induced by DNA repair of top strand lesions of FM1. Dot plots of the fraction of total mutations contributed by each base (squares) and to which base it was mutated (mutational fate, circles) induced by the restoration of T/G (panel **A**) or U/G (panel **B**) to C/G on the top strand of FM1 (pooled according to the type of the lesions). In cases where the base has mutated to transitions and transversions at the same frequency the circles overlap. The beginnings of the promoter (pro), *supF* and MM regions are indicated. The promoter bases are highlighted in yellow and the MM bases in gray. The *supF* bases that are highlighted with the same colors indicate the complementary stem-encoding regions. The inverted black triangles indicate the locations of the mismatches in the pooled data set (see text). The numbers of sequences and mutations are shown at the top of each panel. As the repair of multiple lesions on a given episome could generate a singly mutated plasmid, the number of mutated sequences could be
*Figure 3. Continued on next page*

*Figure 3. Continued*

less the number of introduced lesions (*Figure 2B*). (**C**) Fraction of bases (in terms of top strand sequence) that were mutated in response to the repair (i.e., either restored to C/G or, with respect to T/G, converted to T/A) of the indicated lesions on the top strand of FM1 (pooled according to the type of lesion). (**D**) Dinucleotide context of the mutated base with respect to its 5′ or 3′ base. Fractions of the mutated base induced by the restoration of top strand lesions as a function of its dinucleotide context are shown (pooled from all the lesions restored to C/G). The faint purple bars show the fractions of all 16 dinucleotides in the reporter region. Means (±SEM) are from eighteen alignments (343 sequences, 478 mutations).

The following figure supplements are available for figure 3:

**Figure supplement 1**. Dinucleotide context of mutations induced by the restoration of the indicated lesion generated by reconstituting the FM1 episome on the top strand with the indicated oligonucleotides.

**Figure supplement 2**. Dot plots of mutations induced by the restoration of bottom strand T/G or U/G on FM1.

**Figure supplement 3**. Mutational spectra induced by the repair of lesions on the bottom strand of FM1.

**Figure supplement 4**. Dinucleotide context of mutations induced by the restoration of bottom strand lesions on FM1.

**Figure supplement 5**. Dinucleotide context of mutations induced by the conversion of top or bottom strand T/G to T/A.

**Figure supplement 6**. Most mutations induced by SSB repair involve the C of repair template TpCs.

**Figure supplement 7**. The 0 MM mutational signature is consistent with the mutational spectra of repairing a mixture of top and bottom strand SSBs.

**Figure supplement 8**. Trinucleotide context of mutated G induced by the restoration of top strand lesions on FM1.

panels of *Figure 2C*). Reversing the orientation of the entire reporter cassette (the DNA between the EcoRI and NheI sites [*Figure 1A*] to produce FM1_R [*Figure 2C*, lower panel]) has the same effect. These results also show that strand-specific mutagenesis is neither due to transcription nor DNA replication effects (i.e., transcribed vs non-transcribed strand, and leading vs lagging strand, e.g., *Fijalkowska et al., 1998*; *Hanawalt and Spivak, 2008*) imposed on the reporter region by the episome backbone.

BER would be recruited to T/G, hmU/G and U/G mismatches (*Robertson, 2009*; *Jacobs and Schar, 2012*). Removal of the mismatched T, hmU, or U by a glycosylase and cleavage of the DNA 5′ of the ensuing (or introduced) abasic site (ab), and modification of the 3′ ends ultimately generates a single strand break (SSB, arrow immediately 5′of the green triangle *Figure 2A*, also see *Figure 4A*) that could be extended in the 5′ to 3′ direction by either a single (short patch) or several (long patch) nucleotides. Neither is error prone nor would BER remove the DNA strand on the 5′ side of the lesion (green triangle, *Figure 2A*), which would expose its complement as the template for the error-prone process that would generate mutations in the reporter region. However, components of the MMR pathway can hijack U/G-BER intermediates and generate such gapped substrates (e.g., *Kadyrov et al., 2006*; *Schanz et al., 2009*; *Pluciennik et al., 2010*; *Peña-Diaz et al., 2012*) and we show later in the paper that components of both BER and MMR are involved in the mutagenesis.

The most right hand bar graphs in the upper and middle panels in *Figure 2C* also show that a preformed SSB (i.e., one did not result from BER activity on the introduced mismatches) can also induce mutagenesis in flanking DNA. Here, we replaced respectively the top or bottom strands of FM1, and bottom or top strands of M1F with non-5′ phosphorylated versions of the 0 MM control oligonucleotide. The preformed SSB <50 bp from the *cis* 3′ end of the reporter region in the top strand of FM1 or bottom strand of M1F generates the same mutagenic effect as U/G-ab/G at these positions. We also obtained this result when using non-5′ phosphorylated versions of T/G-mismatch producing oligonucleotides (*Figure 2—figure supplement 4*). Therefore, processing this nick bypasses T/G repair, and generates a substrate that is susceptible to mutagenesis. However, unlike the other lesions, a preformed SSB at ~5 kb from the *cis* 3′ end of the reporter region (bottom and top strand respectively for FM1 and M1F) has little if any mutagenic effect. This difference could reflect the fact that these SSBs would be substrates for a SSB repair (SSBR) pathway that would recruit proteins different from those at the SSB generated during BER (*Caldecott, 2008*).

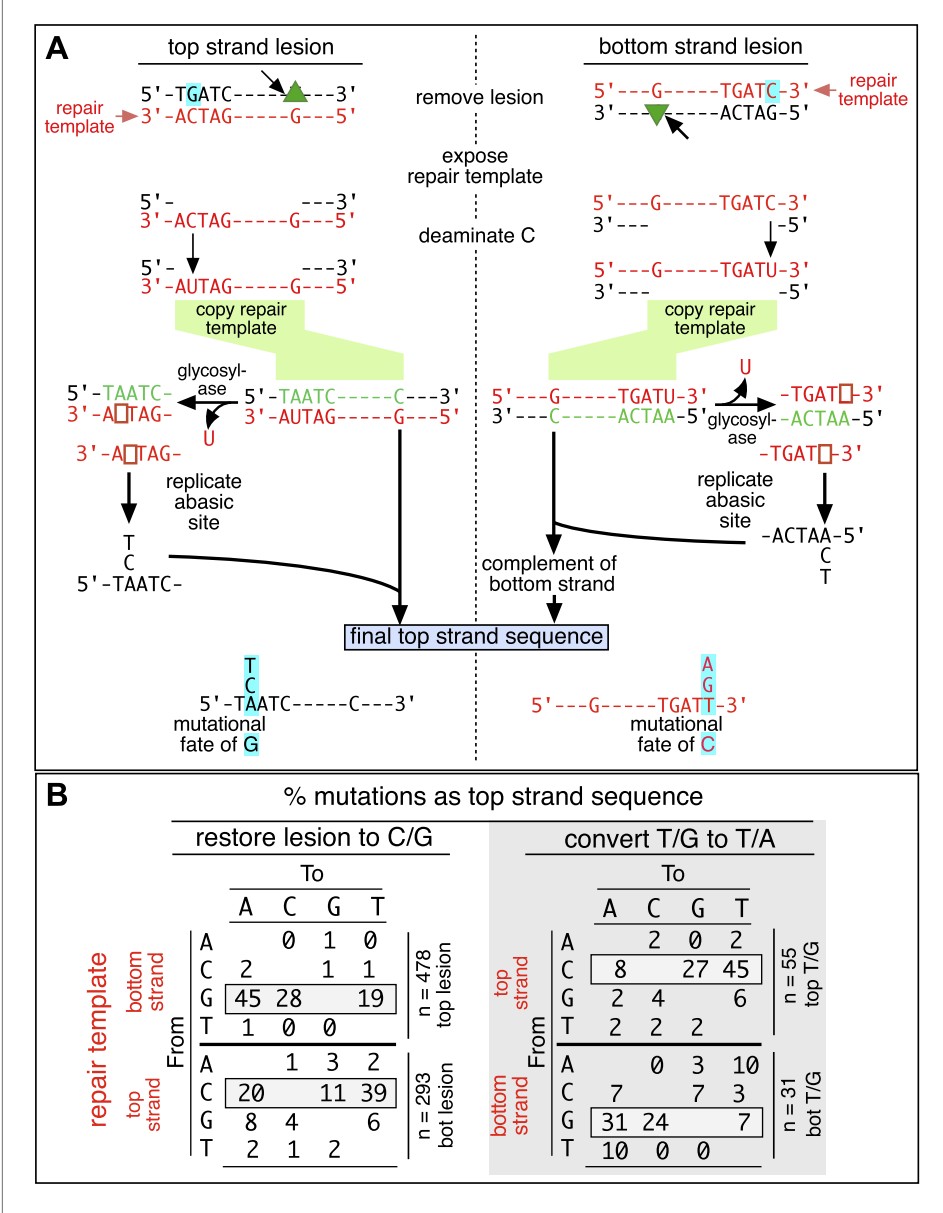

**Figure 4**. Mutagenic process and mutational outcome. (**A**) Graphical representation showing the expected repair template strands (red) for the mutagenic repair of top strand (left panel) or bottom strand (right panel) lesions that were restored to C/G. BER would generate a nick 5′ to the lesions (black arrow). Subsequent exposure of the respective bottom or top strand as a single-stranded repair template would render TpC susceptible to deamination by APOBEC to TpU. The U could be subject to several different processes that would eventually register as a G mutation on top strand GpA (left panel) or as a C mutation on top strand TpC (right panel). See text for more details. (**B**) Identity and fate of the mutated bases (as percentages of the total mutations) in terms of top strand sequence induced by repair that either restored lesions to C/G (left quadrants) or converted T/G to T/A (right quadrants, gray box) for top strand (upper two quadrants) and bottom strand (lower two quadrants) lesions on FM1. The numbers of mutations (n) are given for each category. The data are pooled according to the fate and the strand location of the lesions.

## The mutational spectrum, sequence context, and fate of the bases mutated in response to DNA repair are consistent with APOBEC-mediated mutagenesis

We determined the fraction of total mutations contributed by each base of the reporter and mismatch regions, and the identity of the base to which it was mutated (i.e., its mutational fate), and report these

results in terms of the top strand sequence. (*Supplementary file 2* lists all the mutations for each mutated episome.) The pooled data for top strand T/G or U/G that were restored to C/G are shown in *Figure 3A,B* respectively. The mutations are distributed throughout the cassette, mostly on G, irrespective of the type of lesion (summarized in *Figure 3C*). Mutations of either G or C would equally affect the integrity of the *supF* tRNA stems and potentially produce a non-functional tRNA. Thus, the skewed mutational pattern toward G does not likely reflect an ascertainment bias. In addition, the sequence context of ~80% of the G mutations is GpA, but with less specificity for the 5' base (*Figure 3D*). *Figure 3—figure supplement 1* shows that essentially the same mutational spectrum was induced by restoring each type of top strand lesion—that is, T/G, hmU/G, U/G or ab/G to C/G. Thus a similar mutagenic process is recruited during repair of any of these lesions.

The restoration of top strand lesions to C/G would employ the bottom strand as the repair template. Thus, the preponderance of mutations on the G of GpA indicates that its complement on the repair template, the C of TpC, is targeted by the mutagenic process that is recruited during repair. This preference implicates the TpC-preferring APOBEC family members of single strand-specific cytidine deaminases. The left panel of *Figure 4A* outlines how this could occur during the restoration of top strand lesions to C/G after the repair process was initiated by BER at the lesion (green triangle), and subsequent resection of the lesion-containing strand by MMR as described in the previous section. Deamination of the C of TpC would generate TpU, and as discussed by others (*Lange et al., 2011*; *Nik-Zainal et al., 2012*; *Peña-Diaz et al., 2012*; *Roberts et al., 2012*; *Burns et al., 2013*; *Leonard et al., 2013*; *Roberts et al., 2013*; *Taylor et al., 2013*) further processing of the U could generate a number mutational outcomes of the original C (and its complement G). Faithful copying of the U in the repair template would produce an A/U pair. Subsequent replication would result in a G to A transition (complementary C to T transition). On the other hand, the A/U base pair could also be a substrate for a U glycosylase, which would remove the U and generate an abasic site. Subsequent replication of this strand by a high fidelity DNA polymerase would generally insert an A opposite the abasic site–that is, the 'A rule', (*Strauss, 2002*). The fact that the most frequent mutational outcome at GpA was a G to A transition is consistent with the above two mechanism of generating an A opposite the original C of TpC (*Figure 4B*, upper left quadrant). On the other hand, replication across the abasic site by various other DNA polymerases could insert a T or C opposite the abasic site and generate G to T or C transversions (*Lange et al., 2011*), which we also found (*Figure 4B*, upper left quadrant). As a consequence approximately equal amounts of transitions and transversions result from repair-induced mutagenesis, an outcome that was also observed for APOBEC-mediated mutations in some tumors (*Burns et al., 2013*; *Leonard et al., 2013*). All the data shown in *Figure 4B* are the pooled mutational fates induced by repair of all the lesions, and similar results were found for each type of lesion (results not shown).

The right hand side of *Figure 4A* illustrates the steps when bottom strand lesions are restored to C/G. In this case, the top strand serves as the repair template, but because we show all the mutations in terms of the top strand sequence the final mutational outcome is basically the complement of the results generated from the repair of top strand lesions. *Figure 3—figure supplement 2* shows the fate and mutational spectra (summarized in *Figure 3—figure supplement 3*) induced by repair of bottom strand lesions. Now, most of the mutations are on C of TpC when the lesions are restored to C/G (*Figure 3—figure supplement 4*), and, as the lower left quadrant of *Figure 4B* shows, the most common mutational outcome involves C to T transitions and with about an equal amount of transversions, to A and G.

The right side of *Figure 4B* (gray box) shows the mutational outcome of converting top or bottom strand T/Gs to T/As. In these cases, the top or bottom strand respectively serves as the repair templates (indicated in red font, right side *Figure 4B*), and the mutational outcomes parallel those that found for the restoration of bottom or top strand lesions to C/G wherein the top or bottom strands respectively were used as the repair template. Thus, when a top strand T paired with a bottom strand G is converted to T/A, most of the mutations are on C (*Figure 3C*, T/G_con), ~80% of which involve the C of TpC with almost no specificity for the 3' base (top panel, *Figure 3—figure supplement 5*). However, when a bottom strand T/G is converted to T/A, most mutations are on the G of GpA (*Figure 3—figure supplement 3*–T/G_con, *Figure 3—figure supplement 5*, bottom panel).

Approximately, equal amounts of G and C mutations are generated from the 0 MM controls (most left hand bar graphs, *Figure 3C*, *Figure 3—figure supplement 3*). These mutational spectra and their dinucleotide sequence contexts resemble what would result from repair of a mixture of top and

bottom SSBs (*Figure 3—figure supplement 6*, *Figure 3—figure supplement 7*), and are thus consistent with that expected from repairing trace amount of random nicks on both strands.

APOBEC-mediated mutations in some cancer cells showed a strong preference for the C of TpCpW (W is T or A) (*Roberts et al., 2013*). The trinucleotide context for opposite strand G mutations would be ApGpA and TpGpA. Although the trinucleotide context of about half of our G mutations was ApGpA, the others were mostly CpGpA and not TpGpA, which not surprising as the latter trinucleotide is represented only once in the reporter region (*Figure 3—figure supplement 8*). These results indicate that the APOBECs do not strongly discriminate between the nucleotides that are 3′ to TpC (readily apparent in *Figure 3—figure supplement 4*). A lack of discrimination on 3′ nucleotides was also shown in vitro for APOBEC3B (*Burns et al., 2013*), and reflected in the mutational effect of APOBECs on HIV (*Bishop et al., 2004*; *Doehle et al., 2005*), and in this case the very low preference toward TpCpG (opposite strand CpGpA), reflected the strong bias against CpG in the HIV genome (*van der Kuyl and Berkhout, 2012*).

## Repair-induced mutagenesis can cause clustered mutations

Clustered mutations can be characteristic of APOBEC-mediated TpC (GpA) mutations in tumors (*Nik-Zainal et al., 2012*; *Roberts et al., 2012, 2013*; *Burns et al., 2013*; *Taylor et al., 2013*) and also for AID-induced mutations in lymphoid cells (e.g., *Peled et al., 2008*). *Figure 3* and *Supplementary file 2* showed that the reporter/mismatch region (~300 bp) of some episomes contained more than one mutation. Thus, mutations can occur in a concerted fashion, which is the case for 25–30% of the episomes (*Figure 5A,B*). This figure also shows that the mutational spectra and the dinucleotide contexts of the clustered and unclustered mutations induced by repair of top strand lesions did not differ, and all exhibited the mutational preference for the complement of the C of TpC, that is, the G of GpA. However, the spectra and dinucleotide context of clustered and unclustered mutations induced by repair of bottom strand lesions differed. In contrast to the clustered mutations, which had the same mutational APOBEC signature as the top strand lesions (*Figure 5B*), the unclustered mutations show less of a preference towards mutating the C of TpC, most evident for those induced by restoration of T/G to C/G (cf. lower 2 panels, *Figure 5A* and the lower panels labeled unclustered, *Figure 5B*). Perhaps, the generation of unclustered mutations with a non-strictly TpC mutational signature in the reporter region is facilitated by the ~5 kb distance between its *cis* 3′ end and the lesion (top panel, *Figure 2C*).

## TpC preferring APOBEC C deaminases mediate repair induced mutagenesis

The results in the foregoing two sections strongly implicated the involvement of APOBEC deaminases in repair-mediated mutagenesis. We examined this possibility in two ways: first, after determining the repertoire of APOBEC deaminases in the HeLa cells used for the foregoing experiments, we examined the effect of their knockdown by siRNA on repair-induced mutagenesis. Then, we used ChIP to determine if APOBEC could access the episome in response to an introduced lesion.

### siRNA knockdowns of APOBEC deaminases

qRT-PCR (*Refsland et al., 2010*; *Burns et al., 2013*) showed that several APOBEC deaminases, including A3B, A3F and A3C (*Conticello et al., 2007*; *Prochnow et al., 2009*) are expressed in the HeLa cells (HeLa_JM) that was used for all the experiments presented here unless otherwise stated (*Figure 7A*). A3B, A3F, and A3C exhibit >40% preference for TpC (*Bishop et al., 2004*; *Langlois et al., 2005*), and *Figure 6A* shows that siRNA directed at them reduced their expression. While knockdown of any one of them marginally affected the mutational frequency, knockdown of two (A3B and A3F) inhibited mutagenesis associated with T/G and U/G repair (middle panel, *Figure 6A*). Although knocking down all three did not further reduce the mutagenic effect, the lower part of *Figure 6A* shows that only knocking down all three deaminases (siA3(B,F,C)) eliminates the mutational signature associated with deamination of the C of TpC. However, even though the single knockdowns did not substantially inhibit the mutational frequency, they do change the mutational signature with respect of the dinucleotide context (hereafter referred to as the dinucleotide signature) consistent with the extent of their preferences for TpC (*Bishop et al., 2004*; *Langlois et al., 2005*). Thus, knocking down A3C (46% preference for TpC) hardly alters the dinucleotide signature, whereas knocking down A3B (91% preference) had a more profound effect, and the knockdown of A3F (intermediate at 77%) produced an intermediate effect on this signature. Although the numbers of mutations analyzed are relatively low,

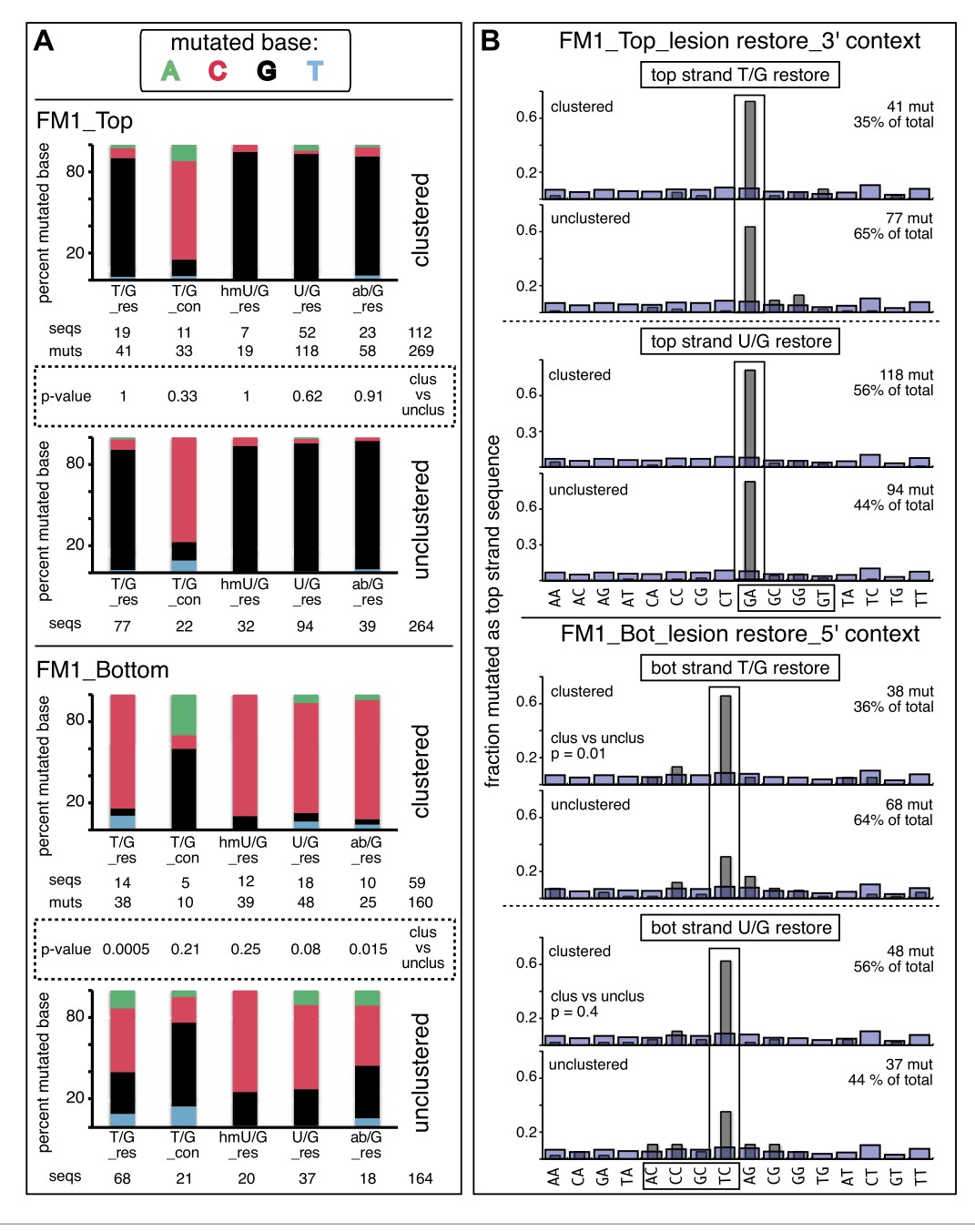

**Figure 5**. Mutational spectra and dinucleotide context of clustered and unclustered mutations. (**A**) Fraction of the bases (in terms of top strand sequence) that were mutated in response to the repair of the indicated lesions on the top (top two panels) or bottom (bottom two panels) strand of FM1 for the clustered (more than one mutations per sequence) or unclustered (one mutation per sequence). The numbers of the mutated sequences and the mutations are shown below each bar graph. p-values were calculated using the Fisher exact test. (**B**) Dinucleotide context of the clustered and unclustered mutations with respect to its 5′ or 3′ base. The faint purple bars show the fractions of all sixteen dinucleotides in the reporter region. The number and percentage of clustered and unclustered mutations are shown. p-values were calculated using the Fisher exact test. The data are pooled according to the type and strand location of the lesion.

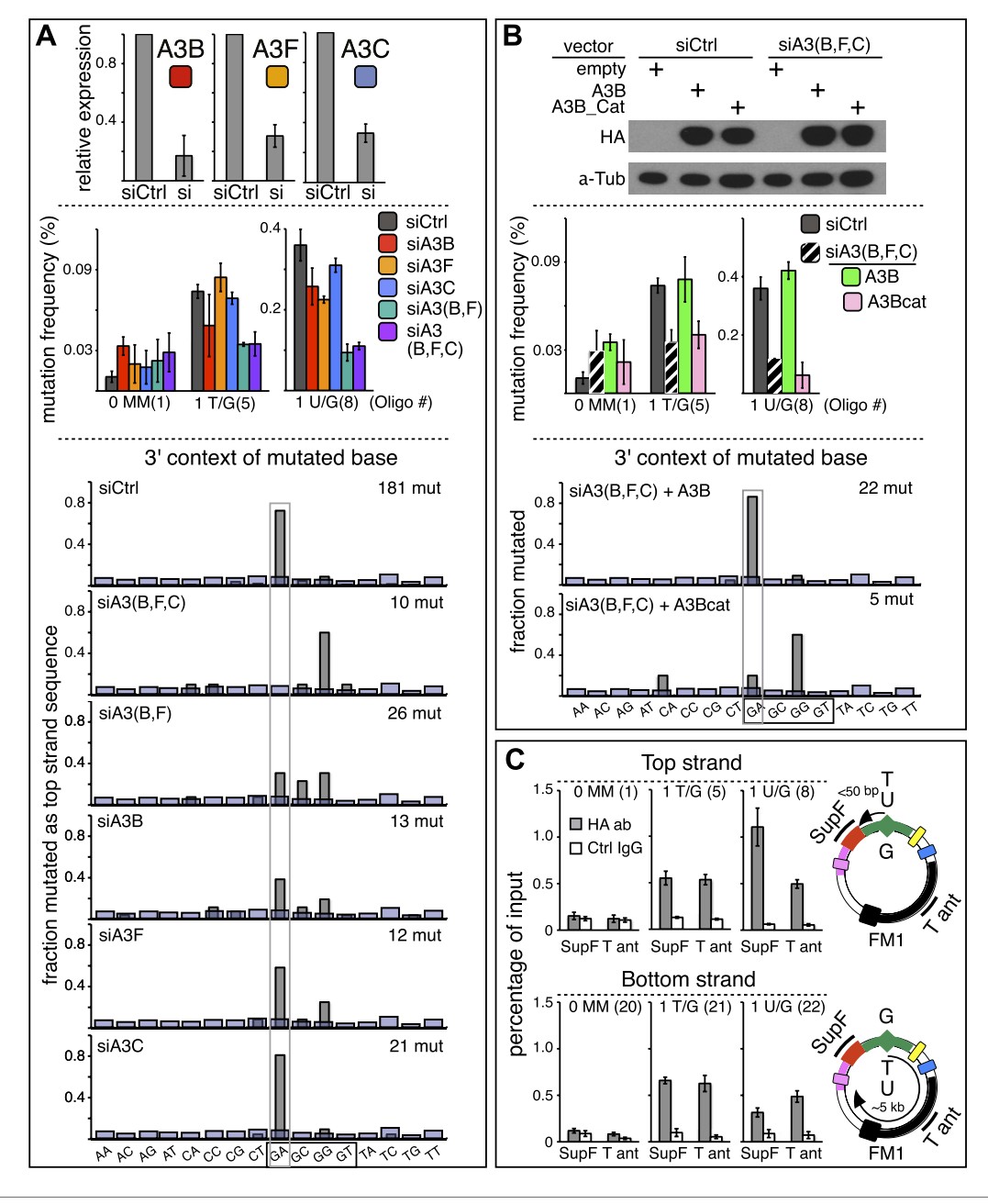

**Figure 6**. APOBEC deaminases mediate repair-induced mutagenesis. (**A**) Effect of APOBEC knockdowns. Top panel, the effect of siRNA knockdown on expression of the TpC preferring, A3B, A3F and A3C deaminases as measured by qRT-PCR. Middle panel, the mutation frequency plotted as a function of the indicated lesion on the top strand of FM1 for the cells transfected with siCtrl (control siRNA), siA3B, siA3F, siA3C, siA3B and siA3F, or all three. Means (±SEM) are shown from at least two independent experiments. The number of oligonucleotide (Oligo #) used to reconstitute the shuttle vector is given below each bar graph and the sequences of these lesion-containing oligonucleotides are listed in **Supplementary file 1**. Bottom panel: the dinucleotide spectra are derived from pooled data of the mutated bases induced by restoration of T/G and U/G on the top strand of FM1 in the presence of control siRNA (siCtrl) or the indicated APOBEC siRNAs. The faint purple bars show the fractions of all sixteen dinucleotides in the reporter region. (**B**) Rescue of APOBEC deaminase knockdown. Top panel, western blots with anti-HA antibody show the synthesis of the siRNA resistant A3B (carboxy-terminal tagged with the HA antigen) or its catalytically inactive version (A3B_Cat) in the presence of control siRNA (siCtrl) or siRNAs against A3B and A3F and A3C–siA3(B,F,C). α-Tub (α-Tubulin, loading control). The middle panel shows the effects of these conditions on
*Figure 6. Continued on next page*

*Figure 6. Continued*
the mutation frequency induced by repair of the indicated lesions—the cross hatched bar graphs show the corresponding results (i.e., siA3(B,F,C) – dark purple bars) from panel **A**. Means (±SEM) are shown from at least two independent experiments. Bottom panel, the 3′ dinucleotide context of the mutated base induced by restoration of the lesions (pooled from the T/G and U/G mismatches) in the presence of siRNA resistant A3B and A3B_Cat deaminases. (**C**) ChIP-qPCR assays to detect the binding of A3B-3HA to the FM1 episome using anti-HA antibody (gray bar) or a control antibody, anti-FLAG IgG (open bar). Diagrams on the right indicate the regions of FM1 that are cognate to the qPCR primers, and the distances between the lesion and the *cis* 3′ end of the reporter. Bar graphs show enrichment of the DNA fragments pulled down by the antibodies for 0 MM-, T/G- or U/G-containing episomes. Means (±SEM) are shown from two independent experiments.
The following figure supplements are available for figure 6:

**Figure supplement 1**. Trinucleotide context of the mutated C of TpC induced by DNA repair after APOBEC knockdown read out as top strand G mutations.

these data suggest that while the three deaminases compete for substrates generated during repair, A3B is the most important contributor to repair-induced mutagenesis.

The results in *Figure 6B* support a predominant role for A3B. Expression of hemagglutinin (HA)-tagged A3B, which is resistant to siA3B (top panel, see 'Materials and methods') reversed the mutagenic effect of the triple knockdown of A3B, A3F and A3C (siA3(B,F,C), middle panel), and restored the TpC mutational signature, but not if it lacked deaminase activity, A3B_Cat (bottom panel, *Figure 6B* and bottom two bar graphs, *Figure 6—figure supplement 1*). This result corroborates the requirement for the deaminase activity. Although we only recovered five mutants in the latter experiment—due to the greatly reduced mutagenic activity, the results agree with those of the triple knockdown shown for the ten mutations shown for siA3(B,F,C) in *Figure 6A*.

## APOBEC3B accesses the SV40 episome in a lesion-specific way

We carried out ChIP to directly determine if the A3B deaminase can access the lesion-containing episome. *Figure 6C* shows that A3B binds T/G- or U/G-containing FM1 episomal DNA in parallel with their relative mutagenic effects and their top or bottom strand location. Thus, more deaminase is bound to the *supF* reporter of top strand U/G- than bottom strand U/G-containing episomes (cf. mutational frequencies in *Figure 2*, panel C). Also, as predicted from *Figure 2C*, the extent of deaminase binding to the *supF* reporter of T/G-containing episomes is indifferent to the top or bottom strand location of the mismatch. *Figure 6C* also shows that the deaminase binds to other episomal regions, for example, the T antigen-containing region for both mismatches. This would be expected as the mutagenic effect can extend over 5 kb for both lesions, which is undiminished for T/G but not for U/G. Consistent with this difference, the deaminase binds both *supF*- and T antigen-containing regions to about the same extent for either the top or bottom strand location of T/G. However, considerably more deaminase binds to the *supF* reporter gene than T antigen gene for top strand U/G wherein the U is closer to the *cis* 3′ end of the *supF* reporter gene than to the T antigen gene (*Figure 6C*, top panel). On the other hand, somewhat more deaminase binds to the T antigen gene than *supF* for bottom strand U/G wherein the U is closer to T antigen gene than to *supF* gene (*Figure 6C*, bottom panel).

Although these results show that A3B, A3F and A3C are necessary for repair-induced mutagenesis, their presence might not be sufficient. We also screened other cell lines, including a second HeLa cell isolate, HeLa_KU, for their contents of APOBEC family members as well as the extent to which T/G and U/G repair are mutagenic (*Figure 7*). Two of the cell lines (HeLa_KU and 2102ep) exhibited distinct dinucleotide mutational signatures (*Figure 7C*). Also the repertoire of APOBEC enzymes in these cell lines differed, even quite dramatically between HeLa_JM and HeLa_KU (*Figure 7A*). Nonetheless, all the cell lines exhibited roughly similar levels of T/G-induced mutagenesis: about the same for HeLa_JM, 143b, and HeLa_KU, and about one-half as much for the other three cell lines (*Figure 7B*). In contrast, differences between the extent of U/G repair-associated mutagenesis were more marked. Thus, U/G repair was about fivefold more mutagenic in HeLa_JM than in HEK293 although their APOBEC repertoires were comparable. Therefore, factors other than just the availability of these deaminases are required for repair-induced mutagenesis, in agreement with observations on some cancer cells (*Roberts et al., 2013*).

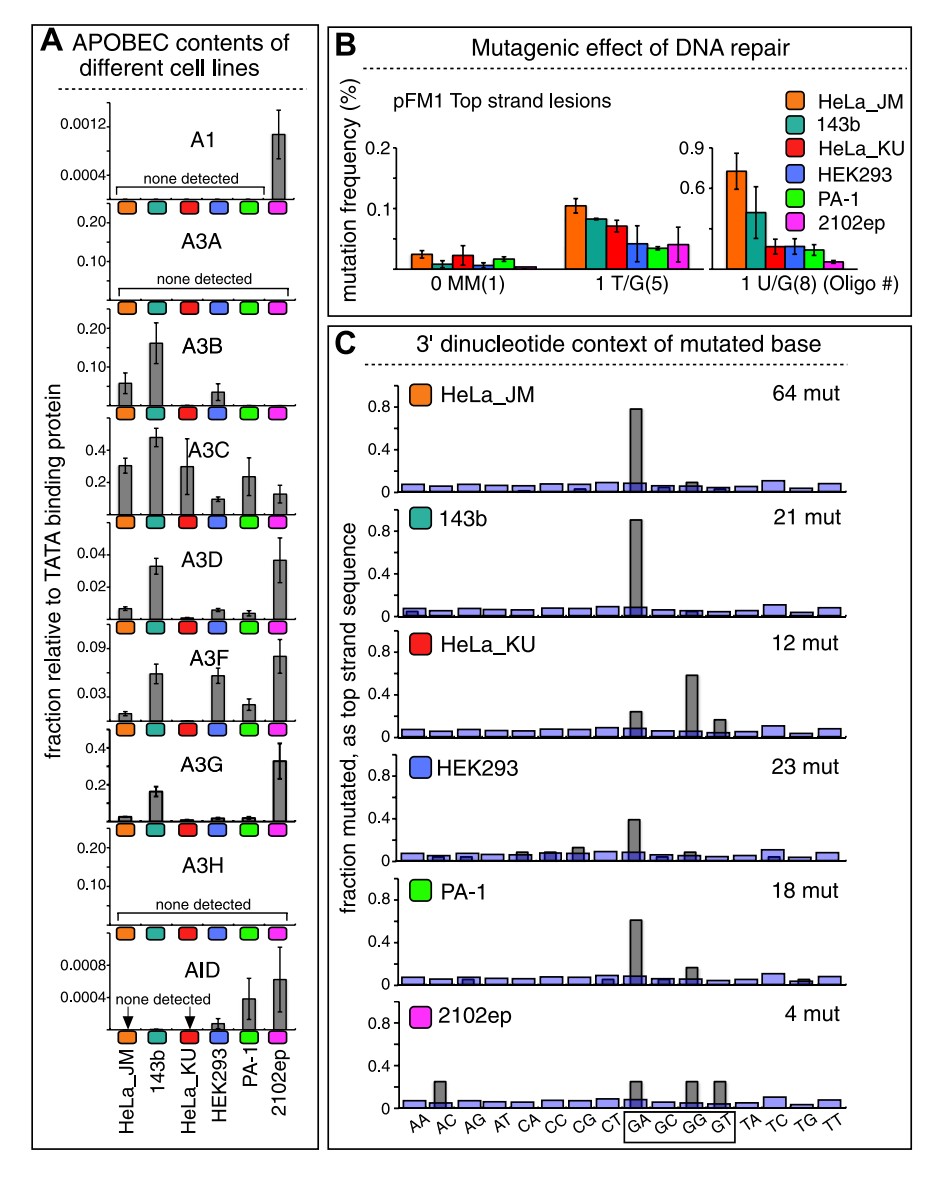

**Figure 7**. The APOBEC content and mutagenic effect of DNA repair in various cells. (**A**) APOBEC contents of different cell lines. APOBEC mRNA expression relative to the constitutively expressed TATA binding protein (TBP) in different cells determined by qRT-PCR. Means (±SEM) are shown from three independent experiments. (**B**) The mutagenic effect of DNA repair. The mutation frequencies as a function of the indicated lesion on the top strand of pFM1 in different cells are shown. Means (±SEM) are shown from at least two independent experiments and the number of oligonucleotide (Oligo #) used to reconstitute the shuttle vector is given below each bar graph. (**C**) The 3′ dinucleotide context of the mutated base induced by restoration of the lesions (pooled from the T/G and U/G mismatches) in different cells. The faint purple bars show the fractions of all 16 dinucleotides in the reporter region.

## siRNA knockdowns implicate the involvement of DNA repair pathways in generating the APOBEC deaminase substrate

As APOBEC-mediated mutagenesis occurred in response to DNA repair, we used siRNA to knock down selected components of different DNA repair pathways and two proteins that would be expected to process or respond to the lesions that we introduced. For these experiments, we treated the reconstituted episomes briefly with the 5′–3′ exonuclease⁻ Klenow polymerase just before transfection into the mammalian cells to remove traces of gapped plasmids (see *Figure 8—figure supplement 1*, the legends to *Figures 8 and 9*, 'Materials and methods').

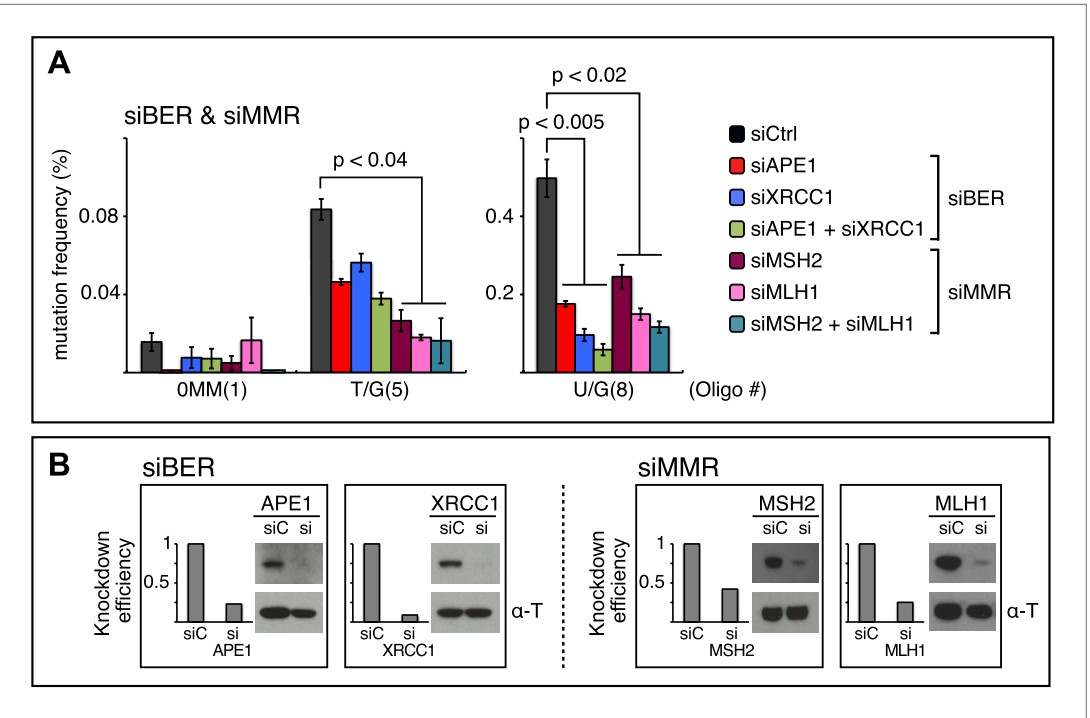

**Figure 8**. siRNA knockdowns of DNA repair pathways reduce mutagenesis. (**A**) Effect of the knockdowns of BER or MMR proteins. The mutation frequency is plotted as a function of the indicated lesion on the top strand of FM1 for control siRNA (siCtrl) and the indicated siRNA transfections. Means (±SEM) are shown from at least two independent experiments and the number of oligonucleotide (Oligo #) used to reconstitute the shuttle vector is given below each bar graph. p-values were calculated using the Fisher exact test. These experiments were carried out with Klenow treated reconstituted episomes (see text and 'Materials and methods'). (**B**) Efficiency of DNA repair protein knockdowns. Western blots for the whole-cell lysates of HeLa cells transfected with siRNAs against various components of the BER and MMR pathways were performed with the antibodies against these proteins and α-T (α-Tubulin, loading control). The western blot results were quantified with Quantity One software (Bio-Rad, Hercules, CA) and normalized to the amount of α-Tubulin to calculate the efficiency of the siRNA knockdowns.

The following figure supplements are available for figure 8:

**Figure supplement 1**. Klenow treatment reduces the gapped contaminants to undetectable levels.

**Figure supplement 2**. Knockdown of DNA repair proteins does not affect the mutational signature.

## BER and MMR

T/G and U/G mispairs are substrates for BER, and BER-repair intermediates generated from U/G were shown to be hijacked by MMR. This process could expose a single stranded template 5′ of the lesion (**Figures 4A and 10**; **Kadyrov et al., 2006**; **Schanz et al., 2009**; **Pluciennik et al., 2010**; **Peña-Diaz et al., 2012**), which could be accessed by the deaminase. Thus, both pathways would be expected to be involved in repair-induced mutagenesis.

The first step in BER involves a glycosylase (**Figure 10**). However, U/G is a substrate for four glycosylases (UNG, SMUG, TDG, MBD4), and T/G for two (TDG, MBD4) (e.g., **Cortázar et al., 2007**; **Robertson, 2009**; **Jacobs and Schar, 2012**). Therefore, we knocked down two BER proteins (APE1 and XRRC1, **Figure 10**) that act downstream of the glycosylases. We also knocked down a component of the heterodimers (MutSα, MutLα) that initiate MMR (MSH2 and MLH1, **Figure 10**). **Figure 8** shows that the knockdown of any one of the above proteins was sufficient to reduce mutagenesis induced by repair of T/G or U/G. While the maximal effect of the knockdowns of U/G-induced mutagenesis was about the same for the BER or MMR, the maximal inhibition of T/G-induced mutation by MMR knockdown was greater than that attained by the BER knockdown. This result is consistent with a model whereby MMR can not only highjack BER intermediates generated from a T/G mismatch but also directly access the mismatch, that is, independently of BER.

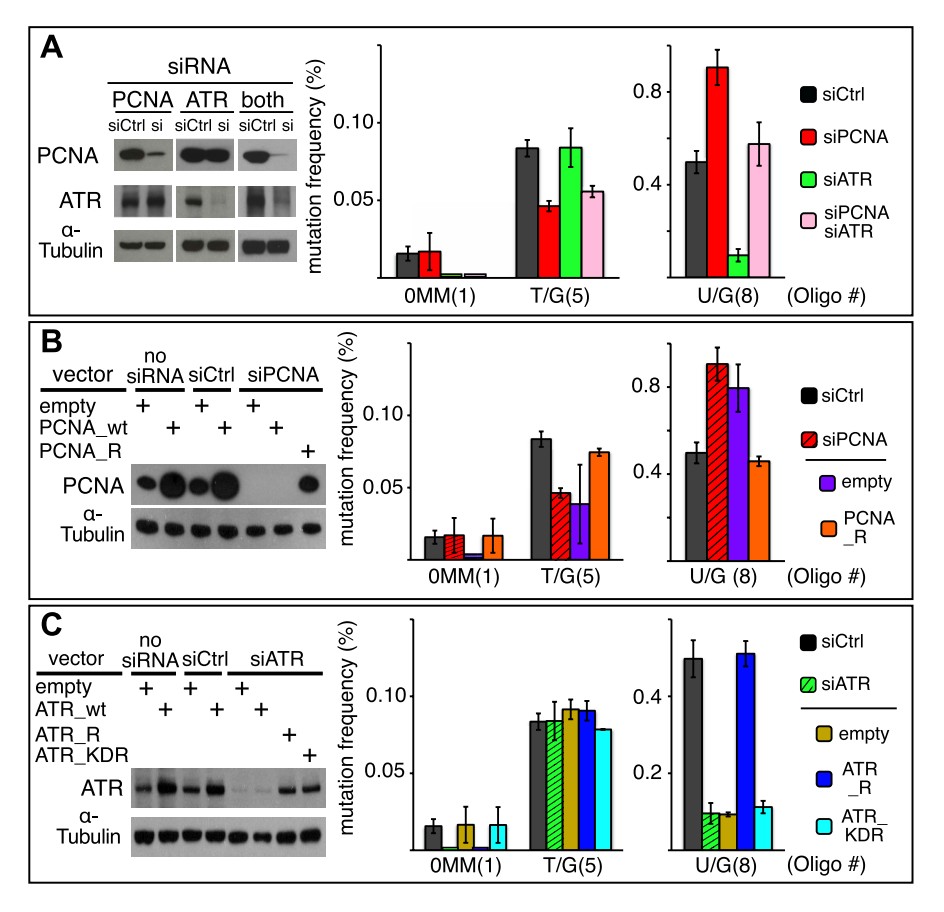

**Figure 9**. Effect of PCNA and ATR on repair-induced mutagenesis. (**A**) Effect of PCNA and ATR knockdowns. The left panel shows the efficiency of PCNA, ATR (or both) knockdowns. The right panel shows the effect of these knockdowns on mutation frequency induced by the repair of each lesion. (**B**) Rescue of the PCNA knockdown. The left panel shows the expression of endogenous PCNA (empty vector), or exogenously expressed wild type PCNA (PCNA_wt), or siRNA-resistant PCNA (PCNA_R) in the absence of siRNA, or in the presence of control siRNA (siCtrl), or siRNA against PCNA (siPCNA). The right panel shows the effects of these conditions on the mutation frequency induced by the repair of the indicated lesions—the cross hatched bar graphs show the corresponding results from panel **A**. (**C**) Rescue of the ATR knockdown. The left panel shows the expression of endogenous ATR (empty vector), or exogenously expressed wild type ATR (ATR_wt), or siRNA-resistant ATR (ATR_R), or siRNA-resistant-kinase defective ATR (ATR_KDR) in the absence of siRNA, or in the presence of control siRNA (siCtrl) or siRNA against ATR (siATR). The right panel shows the effects of these conditions on the mutation frequency induced by the repair of each lesion and the cross hatched bar graphs show the corresponding results from panel **A**. For all panels means (±SEM) are shown from at least two independent experiments and the number of oligonucleotide (Oligo #) used to reconstitute the shuttle vector is given below each bar graph. These experiments were carried out with Klenow treated reconstituted episomes (see text and 'Materials and methods').

The following figure supplements are available for figure 9:

**Figure supplement 1**. Knockdown of PCNA or ATR does not affect the mutational signature.

Direct access of T/G by MMR has been demonstrated in vitro (***Pluciennik et al., 2010***; ***Peña-Diaz et al., 2012***). The knockdown of neither BER nor MMR affected the mutational signature of induced mutagenesis (***Figure 8—figure supplement 2***). Thus, these pathways act upstream of the APOBEC deaminases.

## PCNA and ATR
PCNA and ATR have various roles in DNA metabolism. They can interact with both BER and MMR proteins and the DNA products generated by these pathways (***Moldovan et al., 2007***; ***Schanz et al.,***

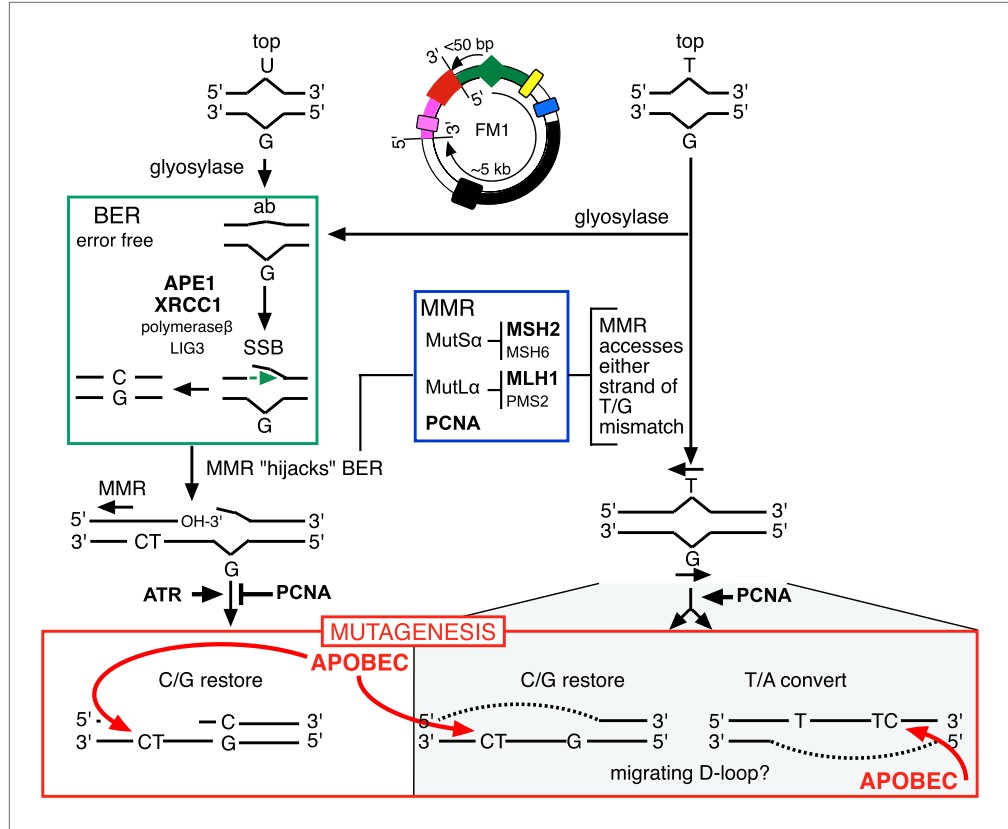

**Figure 10**. Model for APOBEC-mediated mutagenesis induced by U/G or T/G repair. Glycosylases generate the substrate for the subsequent steps of the BER pathway (green box). Cleavage of the DNA 5' of the ensuing abasic site (ab) by APE1, and modification of the 3' ends would generate a single strand break (SSB) that could be extended in the 5' to 3' direction (green arrow) in a reaction involving XRCC1, polymerase-β and LIG3. The blue box encloses components of the MMR pathway. In non-canonical MMR (MutSα and MutLα/PCNA) can hijack U/G-BER intermediates and recruit EXO1 that could generate a gapped DNA 5' of the lesion (see text). The ensuing single strand, which could serve as the template for DNA repair, can be appropriated by APOBEC deaminases (red box) that could deaminate TpC to TpU and processing the U as described in the text and illustrated ***Figure 4A***. As U/G is almost always restored to C/G, the G-strand is the repair template. The right side of the figure shows direct access of a T/G mismatch by MMR (see text). In this case, either the G-strand or T-strand can serve as the repair template for T/G repair, and the T/G-induced mutagenic effect can propagate undiminished for the 5-kb length of the episome. Therefore, we depict the T/G repair intermediate as a migrating D-loop. We show ATR and PCNA acting just before the APOBEC deaminase because the only evidence that we have for the site(s) of action of these factors is somewhere upstream of the APOBEC step (see 'Discussion' for more details). The repair factors that we knocked down are in uppercase, bold typeface.

*2009*; ***Pluciennik et al., 2010***). PCNA acts as a DNA replication clamp that is involved in numerous aspects of DNA replication and can also be involved in both generating gapped substrates and assembling DNA replication complexes on them (***Andersen et al., 2008***; ***Caldecott, 2008***). ATR binds to single-stranded gaps whereupon its kinase activity is activated and phosphorylates various cellular factors (***Liu et al., 2010***; ***Flynn and Zou, 2011***; ***Nam and Cortez, 2011***; ***Pabla et al., 2011***).

***Figure 9A*** shows that the knockdowns of PCNA and ATR have different effects on T/G and U/G repair-induced mutagenesis. Inhibiting PCNA synthesis somewhat reduced T/G-induced mutagenesis, but ATR knockdown had no effect. In contrast, PCNA knockdown stimulated the mutagenic effect of U/G repair, but ATR knockdown inhibited its effect. Simultaneous knockdown of both proteins returns the mutagenic effect of U/G repair to control levels, a result that suggests the effects of both proteins converge on a common pathway. However, the double knockdowns did not alter the inhibition of T/G-induced mutagenesis by PCNA knockdown, not unexpected as T/G repair-induced mutagenesis is indifferent to ATR. Panels B and C show that siRNA-resistant versions of these genes, *PCNA_R* or

*ATR_R* rescued the effects of the knockdowns. For comparison, the PCNA and ATR results from *Figure 9A* are shown respectively in panels B and C as cross-hatched bar graphs. *Figure 9C* shows that the kinase activity of ATR is essential for its effect on U/G-repair-induced mutagenesis and further corroborates the siATR knockdown, because ATR deficient in kinase activity does not rescue it.

The difference between the effects of the PCNA and ATR knockdowns on the mutagenic effect of T/G and U/G repair further differentiates their repair-intermediates as was revealed by the results shown in *Figures 2, 6C, 7B and 8A*. Neither PCNA nor ATR knockdown altered the mutational signature of repair-induced mutagenesis (*Figure 9—figure supplement 1*). Thus, similar to the BER and MMR proteins, PCNA and ATR act upstream of the APOBEC deaminases. Although our results do not provide any mechanistic insights on the roles of PCNA and ATR on APOBEC-mediated mutagenesis, they do indicate the levels of, or access to, APOBEC substrates are sensitive to the activities of these proteins.

## Discussion

Here, we showed that repair of normally occurring mismatches (i.e., those that arise from the inherent chemical instability of DNA) can be mutagenic to flanking DNA. *Figure 10* summarizes our results, which suggest that APOBEC-mediated mutagenesis induced by DNA repair acts downstream of the DNA repair pathways that generate the obligatory single-stranded substrates for TpC preferring APOBEC deaminases (*Conticello et al., 2007*). siRNA knockdowns (bold font depicts the factors that we tested) indicate that components of both BER and MMR are required to generate the deaminase substrates from both T/G and U/G, and that PCNA and ATR could modulate the availability of these substrates to the deaminases. With respect to U/G and T/G, we depict the interaction of the BER and MMR pathways in the framework of that proposed from elegant biochemical studies that showed that MMR can hijack the 3′ end of the nick that would result from a U/G-BER intermediate and generate a single stranded region 5′ of the lesion that could serve as an APOBEC substrate (*Kadyrov et al., 2006*; *Schanz et al., 2009*; *Pluciennik et al., 2010*; *Peña-Diaz et al., 2012*).

The cited in vitro studies from the Modrich group also showed that MMR could directly access T/G mismatches in a PCNA-dependent reaction, which we illustrate on the right hand side of *Figure 10*. In such cases, MMR could load on either strand of the T/G containing episome and lead to resection of either the bottom or top strand of the duplex. However, we can only currently score mutagenic events that occur on the 5′ side of the T/G wherein resection of the top strand would expose the *supF* reporter region. Thus, our results suggest an apparent 3′ bias for the MMR-mediated resection. We are now implementing experiments to examine the induction of mutations 3′ of the T/G.

Although the sensitivity of U/G-induced mutagenesis to the proximity of this lesion to the *cis* 3′ end of the reporter region is consistent with the foregoing biochemical results, some of our findings are at odds with the biochemical model—for example, the apparent inhibitory effect of PCNA on U/G-repair induced mutagenesis (knockdown increases its mutagenic effect, *Figure 9*) vs the requirement for PCNA in the MMR hijacking of BER (*Kadyrov et al., 2006*; *Pluciennik et al., 2010*). While these paradoxical results could be reconciled by considering differences between threshold concentrations of PCNA required for its various functions (e.g., *Pluciennik et al., 2010*), our results do not directly address the mechanistic details of the interaction of PCNA and the U/G-repair generated APOBEC substrate. Furthermore, while we depict PCNA and ATR converging on a common pathway downstream of the hijacked BER intermediate, our only rationale for doing so are the results in *Figure 9A*, which show that their simultaneous knockdown cancels the effects of their separate knockdowns.

Likewise, while the effects of knocking down various BER and MMR components, as well as the ATR and PCNA knockdowns distinguish the T/G- and U/G-repair intermediates, the mechanistic relationships between these pathways or factors, and the properties of the T/G repair intermediate that distinguish it from the U/G intermediate are not clear. The fact that the extent of T/G-induced mutagenesis is more sensitive to the knockdown of MMR than BER suggests that MMR can generate an APOBEC-sensitive substrate independent of (i.e., in addition to) that mediated by BER (right hand side of *Figure 10*). Also, that an APOBEC-vulnerable substrate can be generated from either strand of a T/G mismatch—either restoration of T/G to C/G or conversion to T/A can induce mutagenesis—and that it is sensitive to PCNA knockdown (*Figure 9*) are consistent with studies in vitro that showed that MMR components can access either strand of T/G containing DNA in a PCNA-dependent reaction (*Pluciennik et al., 2010*). Given the relative insensitivity of the mutagenic effect of T/G to its distance from a *cis* 3′ end of reporter region, we tentatively propose that a migrating D-loop can be mounted on either strand of the episome to provide a top or bottom strand deaminase substrate (*Figure 10*).

Numerous recent studies support a role for the TpC preferring APOBEC C deaminases, especially APOBEC3B, in generating large numbers of mutations (i.e., the 'mutator phenotype' [*Bielas et al., 2006*; *Venkatesan et al., 2006*]) that characterize the progression, and perhaps initiation, of cancers (*Nik-Zainal et al., 2012*; *Roberts et al., 2012*; *Stephens et al., 2012*; *Burns et al., 2013*; *Leonard et al., 2013*; *Roberts et al., 2013*; *Taylor et al., 2013*). Less certain is the source of the single-stranded DNAs that are the preferred substrates for the APOBEC deaminases. Here, we directly showed that A3B can access the T/G and U/G repair intermediates generated in our mammalian model system (*Figure 6C*) and generate the mutational signatures induced in some cancers (*Figure 6B*, *Figure 6—figure supplement 1*). The hundreds of T/Gs and U/Gs that are generated daily by spontaneous hydrolytic deamination respectively of methyl-C and C would provide a continual source of potential APOBEC substrates. This source could also be potentially augmented by the thousands of other BER-processed lesions that arise daily due to additional spontaneous degradative processes (*Atamna et al., 2000*; *Barnes and Lindahl, 2004*).

Although our findings showed that DNA repair of T/G, U/G, and other lesions can generate APOBEC deaminase substrates, the composition of the repair intermediates that both renders them susceptible to APOBEC-mediated deamination and accounts for their distinct mutagenic properties are unknown. Components of both the BER and MMR pathways that are normally recruited to these lesions can also engage in complex interactions with factors involved in both DNA metabolism and other cellular functions (e.g.,*Cortázar et al., 2007*; *Kovtun and McMurray, 2007*; *Jacobs and Schar, 2012*; *Peña-Diaz and Jiricny, 2012*; *Peña-Diaz et al., 2012*). Additionally, the availability of APOBEC deaminases per se may not be sufficient to support repair-induced mutagenesis (*Figure 7*, also observed in certain cancers, *Roberts et al., 2013*). Thus some of the factors required to either generate APOBEC substrates from repair-intermediates, or recruit the deaminases to them, might not be present in all cells.

Given the fairly ubiquitous distribution of ABOPBEC deaminases in various cell types and tissues (*Refsland et al., 2010*), determining such factors will be important for evaluating the extent to which DNA repair intermediates can be accessed by APOBEC-mediated mutators, and whether DNA repair could be vulnerable to other error-prone processes as well. Such alternate processes might contribute to the distinctive mutational signatures of the unclustered mutations induced by repair of bottom strand lesions (*Figure 5A,B*). Therefore, an important next step is to characterize the composition of the mutagenic and non-mutagenic repair intermediates that assemble on the various mismatches. An advantage of generating these repair intermediates using preformed mispairs on a defined DNA as done here, is that they could be isolated from cells by selection for either the nucleic acid or a presumed protein component (e.g., *Dèjardin and Kingston, 2009*; *Wu et al., 2011*), or by the sequential application of each. The latter approach should enhance the chances of isolating genuine mutagenic intermediates, a prerequisite for more definitive analysis of their functional and biochemical properties.

Repair-associated mutagenesis might not only contribute to the genetic changes that underlie various disease states (e.g., cancer, as discussed above) but also aging and evolutionary change (as has been earlier suggested, e.g., *Huttley et al., 2000*; *Walser and Furano, 2010*), which in some instances might result from normal physiological processes. For example, we recently found that the repair of 5-carboxy-C/G, an intermediate of the physiological demethylation of methyl-C (*Bhutani et al., 2011*; *Wu and Zhang, 2014*), can also be mutagenic. Thus, the normal cycling of the epigenetic methyl-C mark could contribute to the high mutation rate of regulatory sequences thought to contribute both disease processes (*Maurano et al., 2012*) and evolutionary novelty (*Wittkopp and Kalay, 2012*).

## Materials and methods

### Vector preparation

We constructed various versions of a previously described SV40-based shuttle vector (*Seidman et al., 1985*) as described in detail in the next section. Our major modification was the introduction of a mismatch (MM) region immediately 3′ of the bacterial *supF* tRNA gene to generate the prototypical FM1 vector (*Figure 1A*). *SupF* tRNA suppresses an amber mutation in *lacZ*^amb bacteria (ß-galactosidase minus), which permits blue/white screening to assess the activity of the *supF* gene (loss of activity due to mutations in either its promoter region or coding sequence produces a white colony). We also introduced a bar code of eight random nucleotides to uniquely identify each clone (previously named the signature element, *Parris and Seidman, 1992*). In our initial experiments some episomal plasmids suffered large deletions that included all or part of the reporter cassette, which we found was due to over zealous attempts to remove episomes that lack the bar code by digestion with BstZ17I, which is

unique to this region. However, we found that this step was not necessary and eliminating it largely eliminated the deletions (*Figure 2—figure supplement 1*). Nonetheless, all of the mutational frequencies reported here are the percentage of white colonies containing undeleted plasmids—'Materials and methods—Data acquisition and analysis'. The design of the MM region follows the method of *Hou et al. (2007)*, which facilitates removal and replacement of the top or bottom strand of a DNA duplex and is illustrated for the top strand in *Figure 1B* and *Figure 1—figure supplement 1*.

After nicking the MM region with the New England Biolabs (NEB, Ipswich, MA) enzymes, Nt.BbvCI (top strand) or Nb.BbvCI (bottom strand), the nicked strand was removed by incubation with its biotinylated complement followed by removal of the hybrid with streptavidin-magnetic beads. The gapped vectors were purified by extraction with phenol/chloroform and EtOH precipitation. A 100-fold excess of 5′-phosphorylated control (perfect complement, 0 MM) or oligonucleotides that would generate T/G, hmU/G, U/G or ab/G mismatches (lesions) when paired to the gapped vector were added, and the mixture was heated for 5 min at 95°C, then for 4 hr at 40°C, and after slow cooling to room temperature incubated overnight with T4 DNA ligase (NEB). *Supplementary file 1* lists the oligonucleotides and their numbers are indicated in the figures. We monitored the efficiency of gapping and the reconstitution of the gapped plasmids by following the loss and restoration of the KpnI site between the nicking sites (*Figure 1—figure supplement 1*). For those mismatches that would eliminate an AatII restriction site in the mismatch region, we also monitored the reconstitution of the mismatch-containing episomes by their resistance to digestion by this enzyme (*Figure 1—figure supplement 2*, as has been done by others, e.g., *Peña-Diaz et al., 2012*). Using the T4 DNA ligase from NEB, but not from other sources, eliminated the need for purifying closed circular plasmids prior to their use (Walser, J-C and Aschrafi, A, unpublished observations).

However, the reconstituted episomes contained traces of gapped plasmids (*Figure 1—figure supplement 1*), which were inconsequential except when we knocked down various proteins involved in DNA repair (i.e., selected components of the BER and MMR pathways, or PCNA and ATR; see *Figures 8–10*). Most of these knockdowns increased the mutagenic effect of the 0 MM control (and presumably also that of the lesion-containing plasmids, results not shown). We reasoned that perhaps the trace amounts of gapped episomes are not readily repaired when various repair enzymes are depleted and thereby provide ready-made substrates for the APOBEC deaminases. This idea was supported by our finding that knockdowns of any of the tested repair proteins increased the mutagenesis induced by non-reconstituted gapped plasmids (results not shown). A brief treatment of the reconstituted vectors in the ligation reaction with the 5′–3′ exo⁻ Klenow nuclease just prior to their transfection into mammalian cells reduced the amount of gapped plasmids to undetectable amounts (*Figure 8—figure supplement 1*), eliminated the increase of the mutagenic effect of the 0 MM control during siRNA knockdowns of repair proteins, but did not materially affect the extent of the mutagenesis induced by the 0 MM control or the lesion-containing episomes in experiments with the siRNA controls—cf. the siCtrl for 0 MM, T/G or U/G in *Figures 8 and 9* with those in *Figures 2, 6 and 7* (for HeLa_JM). The Klenow treatment consisted of a 10-min incubation at 37°C of 6 µl of the ligation reaction with 3 µl of a solution that contained 0.2 unit of Klenow polymerase (NEB, M0212), 3 × NEBuffer 2 and 100 µM of dNTP.

## Shuttle vectors

### FM1, 2, and 3

pSP189 was digested with BamHI and MluI, and ligated to mismatch (MM) regions 1, 2 or 3, each of which contains a pair of the aforementioned single strand nicking enzyme sites on the top and bottom strands but they differ in G+C content. These were annealed from complementary oligonucleotides: MM1, InsA_BM_t, 5′GATCCCTCCTCAGCTGACGTCGGACGGAGGCGGCGGAACCTAGGTACCTCAG CCCGA3′/InsA_BM_b, 5′CGCGTCGGGCTGAGGTACCTAGGTTCCGCCGCCTCCGTCCGACGTCAGC TGAGGAGG3′; MM2, InsB_BM_t, 5′GATCCATCCTCAGCTGACGTCTAATACGATTATCGATATATAGG TACCTCAGCTTAA3′/InsB_BM_b, 5′CGCGTTAAGCTGAGGTACCTATATATCGATAATCGTATTAGACG TCAGCTGAGGATG3′; MM3, InsC_BM_t, 5′GATCCAGCCTCAGCTGACGTCTCGTACGATGATCGAT CGATAGGTACCTCAGCTGAA3′/InsC_BM_b, 5′CGCGTTCAGCTGAGGTACCTATCGATCGATCATCGT ACGAGACGTCAGCTGAGGCTG3′, to generate respectively the shuttle vectors: pSP189-FM1, pSP189-FM2 and pSP189-FM3.

We then introduced the bar code into these vectors by digesting them with SacI and NheI endonucleases and ligating each to a SacI/NheI-digested PCR fragment that was generated from the template: Sig_T, 5′TAGTACGCGTGAGCTCTANNNNNNNNNTACGTACGGCTAGCAAGCTCAATT3′, with the

following primers: Sig_F, 5'AGCTGAAAGGATGACTAGTACGCGTGAGCTCT3' and Sig_R, 5'CCCGACC TCGACCCGAATTGAGCTTGCTAGCC3'. The random sequence of 8 bp can uniquely identify up to $4^8$ possible members of a plasmid population. These shuttle vectors were called FM1, FM2, and FM3 respectively. Same site mutations that have different bar codes in a given experiment represent independent events. Thus these vectors were never propagated from a single colony but generated anew by inserting the bar code into their parental pSP189-FM1, pSP189-FM2 and pSP189-FM3 vectors respectively.

### M1F

The *supF* gene was amplified from pSP189 with primers (SupF_BM_F, 5'AAAGGATCCTGTTGACAA TTAATCATC3'/SupF_BM_R, 5'AAAACGCGTGGGTATTGAATTTCGGCC3'). The ensuing PCR product was restricted with BamHI and MluI, and ligated to the BamHI/MluI-digested pSP189 to generate pSP189_FF. pSP189_FF was digested with EcoRI and BamHI, and ligated to an MM1 that had been annealed from the complementary oligonucleotides (InsA_EB_t, 5'AATTCCTCCTCAGCTGACGTCGGA CGGAGGCGGCGGAACCTAGGTACCTCAGCCCGG3'/InsA_EB_b, 5'GATCCCGGGCTGAGGTACCTA GGTTCCGCCGCCTCCGTCCGACGTCAGCTGAGGAGG3') to generate the shuttle vector pSP189_M1F. pSP189_M1F was digested with SacI and NheI and ligated to the bar code sequence as we did above for pSP189_FM1 to generate the shuttle vector M1F.

### FM1_R

The entire reporter cassette (*Figure 1A*) was amplified from FM1 with primers (SupF_NheI_F, 5'AAAGCT AGCTGTTGACAATTAATCATC3' and Sig_EcoRI_R, 5'AGAATTCCTAGCCGTACGTA3'), restricted with EcoRI and NheI and ligated to the EcoRI/NheI-digested pSP189 to generate shuttle vector FM1_R.

The structures of all the vectors were verified by DNA sequencing.

## Transfection of mammalian cells, blue/white colony screening, and siRNA knockdown

Cells were seeded in a six-well plate at a density of $3 \times 10^5$ per well. After 24 hr, we added 100 µl of serum-free DMEM that contained 6 µl each of the overnight ligation mixture (1 µg of reconstituted plasmid) and 6 µl of Fugene 6. After 48 hr, the plasmids were extracted from the cells with Wizard Plus SV Miniprep kit (Promega, Madison, WI), digested with DpnI (removes un-replicated input plasmid) and electroporated into *E. coli* MBM7070 (*lacZ*uag_amber), which were grown on LB plates containing 100 µg/ml ampicillin, 1 mM IPTG and 0.03% Bluo-gal (Invitrogen/Life Technologies, Grand Island, NY). After incubation at 37°C overnight and at room temperature for another day (for maximal color development), the plasmids from white colonies were collected and analyzed for deletions by PCR (primer: F4914, 5'CCAGCGTTTCTGGGGTGAGCA3'/R250, 5'TTTTTGTGATGCTCGTCAGG3', which respectively flank the EcoRI and NheI sites that encompass the reporter cassette, *Figure 1A*). The mutation frequency is the number of white colonies that contained undeleted plasmids/total number of colonies. Directly electroporating reconstituted plasmids into *E. coli* generated no white colonies with undeleted vectors (3 deleted vectors /42,000 screened, results not shown). Thus, the mutagenic effect of DNA repair requires passage through mammalian cells.

For siRNA knockdown, we added 20 pmol siRNA (in a total volume of 400 µl serum-free DMEM that contained 6 µl Lipofectamine RNAiMAX, Invitrogen/Life Technologies) to cells immediately after they were plated. To rescue the siRNA knockdowns, we added 1 µg of the siRNA-resistant plasmid in 400 µl serum-free DMEM that contained 3 µl Lipofectamine LTX, Invitrogen/Life Technologies 24 hr after the siRNA was added, and these cells were transfected 24 hr later with the reconstituted plasmids. We changed the media before each addition.

## RNA extraction and quantitative RT-PCR (qRT-PCR)

We extracted total RNA with the PureLink RNA Mini Kit, Ambion/LifeTechnologies according to the provided instructions and synthesized cDNA with ProtoScript II Reverse Transcriptase (NEB). RNA (2.5 µl @ 400 ng/µl) and 1.5 µl random hexamer DNA primers (100 µM) were heated at 65°C for 10 min and immediately put on ice. The reverse transcription mixture containing 4 µl 5 × RT buffer, 2 µl 10 × DTT, 1 µl 10 mM dNTP, 1 µl RNAse inhibitor (SUPERaseIn, 20 U/µl, Ambion/LifeTechnologies), 1 µl ProtoScript II Reverse Transcriptase (200 U/µl) and 7 µl nuclease-free water. The reactions were incubated at 25°C for 5 min, 42°C for 1 hr and then 80°C for 5 min to inactivate the enzyme. The reactions were diluted with 80 µl of nuclease-free water and qPCR was performed as described in *Refsland et al. (2010)* and *Burns et al. (2013)*.

## Preparation of APOBEC3B, PCNA, and ATR expression plasmids

The target site of the human *apobec3b* siRNA is located in the 3' UTR of the native mRNA. Thus, the APOBEC3B expression plasmid, A3B, which contains only the *apobec3b* protein coding sequence, is resistant to *apobec3b* siRNA. A catalytic deficient version of the plasmid, A3B_Cat, was made by introducing the mutations E68A and E255Q (*Burns et al., 2013*). The PCNA gene was amplified from Mammalian Gene Collection—Human (ATCC, Manassas, VA MGC-8367) with primers (PCNA_BAMHI_F, 5'TTCAAAGGATCCCGTTCGCCCGCTCGCTCTGAGGCT3'/OTB7_R, TTTTT GTTTGCAAGCAGCAGATTAC), digested with BamHI and XhoI and then ligated to BamHI/XhoI digested pcDNA3.1 (+) (Invitrogen) to generate the PCNA-expressing plasmid PCNA_WT. The siRNA target site in PCNA_WT was changed to 5'AATAGAAGACGAGGAGGGT3' to generate the RNAi-resistant plasmid, PCNA_R.

The ATR expression plasmids (ATR_WT and ATR_KD) were gifts from Dr Stephen J Elledge (*Cortez et al., 2001*) and the siRNA target site was changed to 5'TACCCGTCTTCTCAGAATAGCTGCA3' to generate RNAi-resistant plasmids ATR_R and ATR_KDR. All site-directed mutagenesis were performed with Agilent, Santa Clara, CA, QuikChange II Kits.

## siRNA

Control siRNA (no target in mammalian cells, R0017) was purchased from Abnova, Walnut, CA. siRNA against *atr* (5'AACGAGACTTCTGCGGATTGCAGCA3') and *msh2* (5'AATCTGCAGAGTGTTGTGCTT AGTA3') were supplied by Invitrogen (Stealth RNAi siRNA duplex). SiRNA against *xrcc1* (M-009394-01-0005), *ape1* (M-010237-00-0005), *mlh1* (J-003906-10-0005), *pcna* (D-003289-10-0005, M-003289-02-0005 for rescue), *apobec3b* (J-017322-08-0005), *apobec3c* (J-013711-08-0002) and *apobec3f* (J-019039-17-0005) were purchased from Dharmacon RNAi Technologies / GE Healthcare, Fairfield, CT.

## Antibodies

Antibodies were purchased from the following sources: antibodies to ATR (ab10312), MSH2 (ab52266)–Abcam, Cambridge, MA; to PCNA (#2586), XRCC1 (#2735), APE1 (#4128), MLH1 (#3515)–Cell Signaling Technology, Boston, MA; to HA (HA-7, H3663); FLAG (Ctrl IgG for ChIP, F7425)–Sigma, St. Louis, MO.

## Chromatin immunoprecipitation (ChIP)

ChIP assays were performed according to a published protocol (*Carey et al., 2009*). HeLa cells were seeded in a 6-cm dish plate at a density of $8 \times 10^5$ and after 24 hr, 1 µg of the A3B-3HA expressing plasmid in 400 µl serum-free DMEM that contained 3 µl Lipofectamine LTX was added. After another 48 hr, 200 µl of solution of serum-free DMEM that contained 12 µl each of the overnight ligation mixture (2 µg of mismatch/DNA lesion-containing plasmids) and 12 µl of Fugene 6 was added to the cells and after 4 hr cells were harvested for ChIP assay. The qPCR primers for detecting the *SupF* region are: *SupF*_F, 5'GGGGCGAAAACTCTCAAGGATCTTACCGCTG3' and *SupF*_R, 5'GGGATCCGGGTATTG AATTTCGGCCGTG3'; for the T antigen region are: 189LT_F, 5'CCAGCCATCCATTCTTCTATGTCAGC AGAGCC3' and 189LT_R, AAGAACAGCCCAGCCACTATAAGTACCATGAA.

## Cell lines

HeLa_JM was provided by Dr John Moran (University of Michigan), and HeLa_KU by Dr Karen Usdin (NIH), 2102ep (a human embryonal carcinoma cell line) was provided by Dr Tom Fanning, HEK293 was provided by Dr Roland Owens (NIH), and PA-1 (an ovarian teratocarcinoma cell line) and 143B (an osteosarcoma cell line) were purchased from ATCC. Unless otherwise indicated all the experiments were carried out with HeLa_JM cells.

## Data acquisition and analysis

Each reconstituted vector was introduced into various cells in two or more independent transfections for a total of ~500 distinct transfections. The replicated plasmids isolated from each transfection were subject to one or more independent trials of blue/white screening. The plasmids from all the white clones were screened for deletions by PCR and all those with undeleted plasmids from a given transfection were sequenced. More than 99% of the undeleted plasmids contained at least one mutation in the reporter region (data not shown). The ratio of white colonies with mutated undeleted plasmids to the total number of colonies examined in the blue/white trials for a given transfection is taken as the mutation frequency.

We aligned the sequences from each transfection to its relevant starting sequence using SEAVIEW (*Galtier et al., 1996*). Overall, we determined the DNA sequences of ~2600 undeleted plasmids, about 5% of which contained small indels in the reporter region that could not be detected by PCR screening. We excluded these sequences from further analysis because their frequency was not correlated with either the type of introduced lesion or even the presence of one. We grouped the rest (~2500) into 155 distinct alignments. We also determined the DNA sequences of ~250 blue clones, grouped into 38 alignments. We parsed these alignments and computed mutational data using custom Unix, Perl, and R scripts (R Foundation for Statistical Computing, http://www.R-project.org). We also used R for statistical analysis as indicated in the legends to the Figures or text. We determined the fate of the introduced DNA lesions, and the frequency with which each base of the reporter and mismatch regions was mutated, and to which base it was mutated (i.e., its mutational fate). We determined the mutagenic effect caused by the repair of each type of lesion (T/G, hmU/G, U/G, ab/G) located on the top or bottom strand from the pooled results of each lesion class irrespective of their location, number, or context within the mismatch region of a given vector as these variables did not materially affect the mutagenic effect of their repair (*Figure 2—figure supplement 2*).

## Acknowledgements

We thank Ms Pritha Bhattacharyya for participating in some of the experiments, Dr Stephen J Elledge for providing the ATR-expressing plasmid, and Drs Jean-Claude Walser and Angelique Aschrafi for their participation in the preliminary implementation of the shuttle vector assay. We also thank Dr Michael Seidman for providing us the pS189 shuttle vector and are most appreciative of his sustained interest and illuminating discussions throughout the course of this work. We also thank Drs Peggy Hsieh and Deborah Hinton for their comments and suggestions regarding this paper. The following reagent was obtained through the NIH AIDS Research and Reference Reagent Program, Division of AIDS, NIAID, NIH: phAPOBEC3B-HA plasmid, Cat #, 11090 from Dr Bryan R Cullen. Financial Disclosure: This research was supported by the Intramural Research Program of the NIH, The National Institute of Diabetes and Digestive and Kidney Diseases (NIDDK). The funders had no role in study design, data collection and analysis, decision to publish, or preparation of the manuscript.

## Additional information

### Funding

| Funder | Author |
| --- | --- |
| Intramural Research Program of the NIH | Anthony V Furano |

The funder had no role in study design, data collection and interpretation, or the decision to submit the work for publication.

### Author contributions

JC, AVF, Conception and design, Acquisition of data, Analysis and interpretation of data, Drafting or revising the article; BFM, Acquisition of data

## Additional files

### Supplementary files

• Supplementary file 1. Oligonucleotides used to reconstitute the shuttle vectors.

• Supplementary file 2. Mutations

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
