## [Decision Letter]

Thank you for sending your work entitled ”Repair of naturally occurring mismatches can induce mutations in flanking DNA” for consideration at *eLife*. Your article has been evaluated by a Senior editor and 3 reviewers, one of whom is a member of our Board of Reviewing Editors.

The Reviewing editor and the other reviewers discussed their comments before reaching this decision, and the Reviewing editor has assembled the following comments to help you prepare a revised submission.

Chen et al. constructed an episome containing SV40 and pBR origins of replication and a 51 bp sequence bearing a mismatch or DNA lesions (T/G, U/G, hmU/G, abasic). Previous work showed that these lesions can form as the result of deamination, demethylation or glycosylase removal of cytosine residues. The vector also contains the *supF* gene adjacent to the 51 bp sequence. This gene serves as a reporter for mutagenesis in *E. coli* because it can suppress an amber mutation present in a lacZ variant.

Different versions of the episome were created in which a DNA lesion(s) was placed in different sequence contexts on the “top” or “bottom” strand of the 51 bp sequence. The resulting vectors were transfected into HeLa cells that were then incubated for 48 hrs, after which the plasmids were extracted, digested with DpnI, and electroporated into *E. coli*. Bacterial transformants were then assessed for blue (functional supF) and white (supF loss of function) phenotypes. In some cases experiments involving the same lesion but present in different contexts/quantities were pooled. Also, vectors recovered from *E. coli* that contained large deletions in the *supF* gene were not analyzed.

The major results were:

1) Vectors containing no mismatches, the lesions described above, or a single strand break (SSB, present at sites where lesions/mismatches were introduced) were examined for mutations in *supF*. The major finding was that both lesions and an SSB induced mutagenesis that appeared directional (for most substrates) with respect to the location of the lesion. Based on these observation the authors propose that lesions induce a 3'–5' directed excision mechanism, followed by APOBEC-mediated deamination of the ssDNA and subsequent repair synthesis. Curiously the SSB caused the highest level of mutagenesis. Most lesion repair resulted in restoration to a C/G sequence, consistent with a BER-type mechanism.

2) DNA sequencing analysis of *supF* mutations was performed, with the mutation spectra showing a signature consistent with APOBEC mutagenesis. In support of this idea, about 30 % of the episomes displayed “mutational showers,” and siRNA treatment of HeLa cells indicated that a triple knockdown of APOBEC A3B, A3F, and A3C (restored by reintroducing A3B) eliminated the APOBEC mutational signature.

3) siRNA knockdowns of BER, MMR and DNA damage response proteins altered the mutagenesis the frequency, but in general not the mutational spectra.

This work involved the development of clever methods to create plasmids with specific DNA lesions and a significant amount of cell manipulations (different cell lines, siRNA knockdowns, ChIP) and molecular analyses.

1) One major concern is that vectors containing an SSB displayed the highest frequency of mutagenesis (Figure 2). However, knockdowns of BER components (Figure 9) showed that single knockouts provided minor, if any, effect on the mutagenesis frequency. Double knockouts of the BER components APE1 and XRCC1 yielded results that were difficult to interpret because the mutagenesis frequency seen in the 0-mismatch control was elevated to fairly high levels. Together these observations call into question the main conclusion that “naturally occurring mismatches can induce mutations in flanking DNA” and that an SSB is an intermediate in a BER-type pathway that is initiating the mutagenic event. I'm convinced that some type of 3’–5' single-stranded excision mechanism initiated at an SSB is occurring that leads to APOBEC deamination of the ssDNA template, followed by resynthesis. However, it is unclear if this is an artifact of the plasmid system or represents a novel in vivo mechanism.

2) We worry about pooling data from a variety of mismatch/lesion substrates (Figure 2). It appears that datasets for individual substrates were too small to analyze statistically and so combining data could skew the analysis. Also, it is not clear to me whether data pooling was performed for work shown in the other figures. This should be made clear. At the very least a single set of vectors should be directly compared in Figure 2 in a large data set.

3) We find it curious that repair in this system appears to be primarily 3' directed whereas repair of G/T mismatches in vitro DNA mismatch repair reactions (Paul Modrich type studies) appears to be bi-directional (nicks are placed 3' or 5' to the mismatch). Can the authors speculate as to the source of the bias?

4) The data presented in Figures 8 and 9 are challenging because in the case of MMR and BER, double knockdowns are required to observe subtle changes in mutation frequency without a change in mutation spectra. Why is it necessary to knockdown two members of the same pathway (i.e., Ape1 and XRCC1)? Similarly, if BER and NER function in the same mutagenesis pathway proposed in Figure 10, why do both need to be knocked down to see an effect? Also, the 0-mismatch substrate shows unusual fluctuations in mutation frequency. For example, there is an unusual mutagenesis pattern in the OMM substrate for the single MLH1 and MSH2 knockdowns compared to the double knockdown (Figure 8) that makes me concerned based on the high mutation rate expected for single knockouts of these critical MMR genes. Unusual patterns are also seen when comparing the single and double BER knockdowns, and the patterns seen for PCNA and ATR are hard to interpret, again because of the fluctuation in the 0-mismatch substrate.

Although there was some disagreement among the reviewers on this issue, it was felt by some that it might be better to remove Figures 8 and 9 from the paper, and publish these results at a later time when the data are more clear or better rationalized.

---

## [Author Response]

We enumerate below our response to each point but start with those raised in points 1 & 4 as we were particularly attentive to the concerns about the interpretation of the results on the effects of inhibiting DNA repair pathways shown in Figures 8 and 9, and that some of these data defied a straightforward interpretation. We also were mindful of these issues; e.g., in the Results section in reference to the data in Figure 8 we had written:

“Somewhat paradoxically, some of the single and double knockdowns increased the mutagenesis associated with the 0 MM top strand control (Figure 8). The dinucleotide signature of these mutations (Figure 8—figure supplement 1) resembled that induced by repair of a top stand lesion (Figure 3). As reconstituted vectors retain traces of gapped episomes (Figure 1—figure supplement 1), perhaps these gapped molecules are not so readily repaired when various repair enzymes are depleted thereby providing ready made substrates for the APOBEC deaminases.”

And in light of the following comment – “Although there was some disagreement among the reviewers on this issue, it was felt by some that it might be better to remove Figures 8 and 9 from the paper, and publish these results at a later time when the data are more clear or better rationalized” – we felt that we should attempt to clarify the repair knockdown data by directly addressing the issue of whether the traces of gapped molecules in our reconstituted plasmids were skewing our results in these experiments as we had suggested above.

Therefore, we first examined the effect of all the repair knockdowns reported in Figures 8 and 9 on the mutagenic effect induced by a non-reconstituted gapped version of the pFM1 episome. As Figure 11 shows, all the knockdowns increased the mutagenic effect of the gapped plasmid, generally by two-fold or more, which would certainly confound the effect of the knockdowns on the reconstituted vectors, all of which could be contaminated by varying amounts of gapped episomes (see Figure 1—figure supplement 1 of our original & current submission). We could reduce the content of gapped contaminants to undetectable levels by briefly treating the reconstituted episomes with the 3’-5’ exo-Klenow polymerase (described in the Materials and Methods of our revised paper), as is shown in Figure 8—figure supplement 1. We then proceeded to examine the effects of BER, MMR, PCNA and ATR knockdowns with reconstituted episomes that were treated with the Klenow polymerase as described in the Material and Methods (last paragraph of Vector preparation) of our revised paper.Author response image 1.

As the new Figure 8 of our revised paper shows, the effects of the knockdowns on the repair pathways are now quite straightforward, and also more informative:

First, the mutagenic effects of the 0MM, T/G and U/G lesions in the absence of repair pathway knockdowns (i.e., the siCtrl values) were not materially affected by Klenow treatment as they are consistent with those observed for the non-Klenow treated versions of these episomes - cf. the siCtrl for 0 MM, T/G or U/G in our new Figures 8 and 9 with those in Figures 2, 6 and 7 (for HeLa_JM).

Second, the knockdowns of any one of the tested members of either the BER or MMR pathways was sufficient to reduce mutagenesis induced by any of the lesions.

Third, while the maximal effect of the knockdowns on U/G induced mutagenesis was about the same for the BER or MMR knockdowns, the maximal inhibition of T/G-induced mutation by MMR knockdown was greater than that attained by the BER knockdown. This result is consistent with a model whereby MMR can both highjack BER intermediates generated from a T/G mismatch but also access a T/G mismatch directly, i.e., independently of BER. We had suggested the direct access of T/G by MMR in our original paper, referencing the work from the Modrich and Jiricny groups in the third paragraph of the Discussion:

“That either the restoration of T/G to C/G or its conversion to T/A can induce mutagenesis, indicates that either strand of a T/G mispair can be accessed by APOBEC. These results are consistent with the in vitro studies, which showed that purified MMR proteins can access either strand of T/G-containing closed circular molecules (34; 62).”

We now explicitly illustrate this possibility in a modified Figure 10 of our revised paper and this issue also speaks to point 3 wherein the reviewers ask “Can the authors speculate as to the source of the bias?”, which we address in more detail below.

The new Figure 9 of our revised paper shows that the ATR and PCNA knockdowns have distinct effects on T/G repair- and U/G repair-induced mutagenesis. The new results also reveal a partial dependence of T/G-induced mutations on PCNA – i.e*.*, PCNA knockdown decreases the mutagenic of T/G, a result which was obscured in our original submission. This finding is consistent with in vitro results from the Modrich group that showed that direct access of MMR for T/G required PCNA.

The new results in Figures 8 and 9 greatly simplified our discussion of the roles of BER and MMR in repair induced mutagenesis and we have modified both the Results and Discussion sections accordingly.

We now address the remaining points not covered by the above:

*1) One major concern is that vectors containing an SSB displayed the highest frequency of mutagenesis (*Figure 2*). However, knockdowns of BER components (*Figure 9*) showed that single knockouts provided minor, if any, effect on the mutagenesis frequency. Double knockouts of the BER components APE1 and XRCC1 yielded results that were difficult to interpret because the mutagenesis frequency seen in the 0-mismatch control was elevated to fairly high levels. Together these observations call into question the main conclusion that “naturally occurring mismatches can induce mutations in flanking DNA” and that an SSB is an intermediate in a BER-type pathway that is initiating the mutagenic event. I'm convinced that some type of 3’–5' single-stranded excision mechanism initiated at an SSB is occurring that leads to APOBEC deamination of the ssDNA template, followed by resynthesis. However, it is unclear if this is an artifact of the plasmid system or represents a novel* in vivo *mechanism*.

It was stated here that “One major concern is that vectors containing an SSB displayed the highest frequency of mutagenesis (Figure 2).” But this result was only found in one case, Figure 2 bottom strand, wherein the mutagenic effect of SSB was marginally higher than ab/G, whereas for the top strand, the SSB effect was marginally lower than ab/G. In these experiments and others not reported, the mutagenic effect of U/G, ab/G and SSB were generally the same, except, as noted on lines:

“However, unlike the other lesions, a preformed SSB at ∼5 kb from the cis 3’ end of the reporter region (bottom and top strand respectively for FM1 and M1F) has little if any mutagenic effect. This difference could reflect the fact that these SSBs would be substrates for a SSB repair (SSBR) pathway that would recruit proteins different from those at the SSB generated during BER (24).”

(The rest of the issues raised in this point are similar to those raised in point 4, which we addressed above.)

*2) We worry about pooling data from a variety of mismatch/lesion substrates (*Figure 2*). It appears that datasets for individual substrates were too small to analyze statistically and so combining data could skew the analysis. Also, it is not clear to me whether data pooling was performed for work shown in the other figures. This should be made clear. At the very least a single set of vectors should be directly compared in*
Figure 2
*in a large data set*.

We feel that some of this worry could have stemmed from our not being precise about what we meant about “pooling” data, and by using this term for two different purposes we elided the distinction between what we did.

(A) We stated that we pooled the mutational effects of repairing each type of lesion. However, our original (and present) Figure 2 does not show pooled data but a dot plot of the % mutational frequency of every trial of repairing each type of lesion. By calculating a mean from each type of lesion regardless of their number or position or context within the mismatch region, we treat each lesion type (i.e*.*, T/G, U/G) as a distinct class despite the inevitable overlap in some of the values for % mutational frequency induced by their repair. This assumption was borne out by the distinctive mutagenic effect and properties that the repair of these lesions exhibited. We have modified our paper to make these distinctions.

(B) We pooled the mutational spectra – i.e., which bases were mutated and their dinucleotide (and trinucleotide) sequence contexts – that were generated in response to the repair of different lesions. We felt that this was warranted because we recovered the same mutational signature regardless of which type of lesion (i.e*.*, T/G, U/G, etc.) was restored to C/G. For example, see Figure 5 wherein the spectra of mutations induced by the repair of T/G and U/G are shown separately. In fact, even in the one case (unclustered bottom strand mutations, Figure 5) where we detected mutational spectra in addition to that generated by the TpC-specific deaminases, the spectra induced by T/G and U/G repair were very similar.

Nonetheless, we think that the reviewers’ suggestion to show the results for a single set of vectors is a good idea and so we modified Figure 2—figure supplement 2 to include a separate statistical analysis comparing the mutagenic effects 2 T/G, 2 hmU/G, and 2 U/G. The oligonucleotides for reconstituting these lesion-containing episomes are “isogenic” in that the lesions are in the same position (and thus context) of the MM region. We also included an additional supplemental figure, Figure 3—figure supplement 1, which shows that there is no difference between the dinucleotide context of the mutations induced by repair of these lesions as well as by the repair of ab/G. In our revised paper we explicitly point out these results and also indicate which results were derived from what kind of pooled data in the figure legends.

*3) We find it curious that repair in this system appears to be primarily 3' directed whereas repair of G/T mismatches* in vitro *DNA mismatch repair reactions (Paul Modrich type studies) appears to be bi-directional (nicks are placed 3' or 5' to the mismatch). Can the authors speculate as to the source of the bias*?

This bias would reflect the choice of which strand of the T/G mismatch that is selected as the repair template (the right hand part of Figure 10). In our case, and in the pioneering experiments from the Jiricny and Taylor laboratories on the fate of T/G lesions introduced into an SV40 virion (Hare and Taylor 1985, Brown and Jiricny 1987), the G-containing strand was chosen ≥ 75% of the time. This bias towards the G-strand is in part due to the T-specific glycosylases (to our knowledge there are none that are specific for the G of a T/G mismatch). But this mechanism probably does not account for all of the G-strand bias, for in other experiments not reported here we found that the sequence context of the mismatch, including the presence of C-methylation can strongly affect the choice of strand selection as repair template. In some cases, the T-strand is used as the repair template more than 50 % of the time and this can occur whether or not repair is mutagenic. These instances quite likely involve direct access of the MMR pathway to the T/G mismatch. Whatever the extent that MMR can compete with BER to directly access a T/G mismatch, a built in bias of MMR towards accessing the T-containing strand would minimize C to T transitions at CpG sites. Even so, transitions at CpG sites represent the most frequent class of SNPs and, with time, CpGs not under selection are converted to TpGs (CpAs). We have modified our paper to more explicitly address this issue and modified Figure 10 to show that the conversion of T/G to T/A results from the mounting of MMR components on the G-containing strand.